



**Water uptake of subpollen aerosol particles: hygroscopic growth, CCN activation, and liquid-liquid phase separation**

Eugene F. Mikhailov[1,2], Mira L, Pöhlker[1], Kathrin Reinmuth-Selzle[1], Sergey S. Vlasenko[2], Ovid. O. Krüger[1], Janine Fröhlich-Nowoisky[1], Christopher Pöhlker[1], Olga A. Ivanova[2], Alexey A. Kiselev[3], Leslie A. Kremper[1], and Ulrich Pöschl[1]

[1]Multiphase Chemistry and Biogeochemistry Departments, Max Planck Institute for Chemistry, 55020 Mainz, Germany
[2]St. Petersburg State University, 7/9 Universitetskaya nab, St. Petersburg, 199034, Russia
[3]Atmospheric Aerosol Research Department, Institute for Meteorology and Climate Research, Karlsruhe Institute of Technology (KIT), Hermann-von-Helmholtz Platz 1, 76344 Eggenstein-Leopoldshafen, Germany.

Correspondence: Eugene F. Mikhailov (eugene.mikhailov@spbu.ru)

**Abstract**

Pollen grains emitted from vegetation can release subpollen particles (SPP) that contribute to the fine fraction of atmospheric aerosols and may act as cloud condensation nuclei (CCN), ice nuclei (IN), or aeroallergens. Here, we investigate and characterize the hygroscopic growth and CCN activation of birch, pine, and rapeseed SPP. A high humidity tandem differential mobility analyzer (HHTDMA) was used to measure particle restructuring and water uptake over a wide range of relative humidity (RH) from 2 % to 99.5 %, and a continuous flow CCN counter was used for size-resolved measurements of CCN activation at supersaturations ($S$) in the range of 0.2 % to 1.2 %. For both, subsaturated and supersaturated conditions, effective hygroscopicity parameters κ, were obtained by Köhler model calculations. Gravimetric and chemical analyses, electron microscopy, and dynamic light scattering measurements were performed to characterize further properties of SPP from aqueous pollen extracts such as chemical composition (starch, proteins, DNA, and inorganic ions) and the hydrodynamic size distribution of water-insoluble material. All investigated SPP samples exhibited a sharp increases of water uptake and κ above ~95 % RH, suggesting a liquid-liquid phase separation (LLPS). The HHTDMA measurements at RH> 95% enable closure between the CCN activation at water vapor supersaturation and hygroscopic growth at subsaturated conditions, which is often not achieved when HTDMA measurements are performed at lower RH where the water uptake and effective hygroscopicity may be limited by the effects of LLPS. Such effects may be important not only for closure between hygroscopic growth and CCN activation but also for the chemical reactivity, allergenic potential, and related health effects of SPP.

## 1 Introduction

The hygroscopic properties of atmospheric aerosols and their ability to act as cloud condensation nuclei forming cloud droplets are crutial for the radiative budget of the Earth's atmosphere (Hänel, 1976; Rader and McMurry, 1986; Pöschl, 2005; McFiggans et al., 2006; Andreae and Rosenfeld, 2008;





Swietlicki et al., 2008; Cheng et al., 2008; Zieger et al., 2013; Rastak et al., 2014, and references therein). The primary parameters, which determine the optical properties, CCN activity, reactivity, and deposition of aerosol particles are their size and composition. Depending on their sources and atmospheric processing (interaction with radiation, gases, and clouds), atmospheric particles consist

of a complex mixture of organic and inorganic chemical components.

Primary biological aerosols (PBA) are a subset of atmospheric particles, which are directly released from the biosphere into the atmosphere. They comprise viable or dead cells, reproductive units, and fragments of organisms (bacteria, fungal spores, viruses, pollen, plant debris, etc.) (Després et al, 2012; Fröhlich-Nowoisky et al., 2016). PBA particles span the entire range of atmospheric

aerosol particle diameters from nanometers to ~100 μm, whereby the lower limit is given by the size of molecular clusters or macromolecules and the upper limit is related to rapid sedimentation. The number and mass concentrations of PBA particles over vegetated regions are typically on the order of $10^4$ m$^{-3}$ and 1 μg m$^{-3}$, respectively and  account for around 30% in urban and rural air (Després et al, 2012; Fröhlich-Nowoisky et al., 2016). Laboratory experiments have shown that primary bioparticles

are efficient CCN and IN and thus are assumed to influence the formation of cloud systems and precipitation (Möhler et al., 2007, Pöschl et al., 2010, Morris et al., 2014; Hoose and Möhler, 2012). Intense precipitation, in turn, can increase the concentrations of IN active bioaerosols in certain ecosystems (Huffman et al., 2013; Prenni et al., 2013). The positive feedback between the concentration of biological aerosol and precipitation is called *bioprecipitation* (Morris et al., 2014). It

has been suggested that rainforests such as the Amazon Basin can act as biogeochemical reactors in which the precipitation induced by PBA materials sustain the production of plants, and hence PBA material, within the ecosystem (Pöschl et al., 2010).

Pollen is a category of bioaerosols that may spread in large quantities. It is the male gametophyte of gymnosperms and angiosperms. Pollen size ranges from 15 μm to 100 μm; its shape,

when dry, is generally oval or spherical. The typical ambient number concentration of the pollen is estimated to be 10-$10^3$ m$^{-3}$ (Després et al., 2012). In the Northern Hemisphere forests during strong pollination events pollen concentration may exceeds $10^4$ m$^{-3}$ (Williams and Després, 2017). Anemophilous (wind-driven) pollen despite their large size can undergo long distance dispersal up to ~ $10^3$ km (Sofiev et al., 2006; Kuparinen et al., 2009).

Water uptake of whole pollen grains at subsaturated conditions was studied by Pope (2010) and Griffiths et al. (2012) using an electrodynamic balance (EDB), as well as by Tang et al. (2019), utilized a commercial vapor sorption analyzer. It was found that mass hygroscopic growth of pollen grains can be approximated by the κ-Köhler equation. For six pollen species studied by EDB the  κ values were in the range of 0.05-0.10 (assuming a dry pollen density of 1 g cm$^{-3}$). These values agree

with the κ range of 0.03-0.06 reported by Tang et al. (2019) based on gravimetric measurements of



six pollen samples at 25 °C.  The EDB measurements by Pope (2010) and Griffiths et al. (2012)  were complemented by  Environment Scanning Electron Microscopy (ESEM) analysis, which showed that at RH < 85% the pollen grains swell internally by capillary effect through the germ apertures (Diehl et al., 2001). At RH > 89 %, the wetting of the pollen surface was observed.  The surface absorption

of water was first detected at the pollen pore sites and proceeded to engulf the pollen grain as a whole. Pollen has been identified as a potential natural source of giant CCN (Griffiths et al., 2012). Many studies have reported a high IN activity of various pollen species and highlight their potential importance for mixed phase clouds in biologically-influenced environments (Diehl et al., 2001, 2002; von Blohn et al., 2005; Chen et al., 2008; Pummer et al., 2012). Although pollen grains are assumed

to account only for a small fraction of IN active on a global scale, their local and regional impact on cloud microphysics could be substantial (Hoose et al., 2010).

In addition to the direct emission of intact pollen grains, the rupture of pollen and subsequent release of fine particles may contribute more substantially to ambient aerosol particle concentration. Fresh pollen grains can rupture at high humidity (Taylor et al., 2002, 2004) and during precipitation

events (Grote et al., 2001, 2003; Hughes et al., 2020). They may also mechanically rupture under turbulent conditions by impact forces (Visez et al., 2015). When pollen grains rupture, they release cytoplasmic material known as subpollen particles (SPP) ranged from several nanometers to about 1 μm (Grote et al., 2003; Taylor et al., 2004). Rupture can occur on open anthers (i.e., at the surface), and SPP can be subsequently dispersed to the atmosphere by disturbances like wind in dry conditions

(Taylor et al., 2004). Regional transport and weather patterns may facilitate the transport of these smaller particles and lead to longer atmospheric residence time (Wozniak et al., 2018; Hughes et al., 2020).

Augustin et al. (2013) and O'Sullivan et al. (2015) have shown that the nano-scale range SPP washed of pollen grains are IN active even with a molecular mass below 1000 kDa. Laboratory

experiments with six fresh pollen samples revealed that SPP ranging 50-200 nm are CCN active in the supersaturation range of 0.81-0.12% (Steiner et al., 2015). Mikhailov et al. (2019) found an almost constant CCN-derived κ value of 0.12 to 0.13 for SPP produced by birch, pine, and rapeseed pollen grains. As shown below, these values are underestimated since they do not account for the irregular shape of the initial dry SPP. Wozniak et al. (2018) developed the first model of atmospheric pollen

grain rupture and implemented the mechanism in regional climate model simulations over the spring pollen season in the United States with a CCN-dependent moisture scheme. They showed that in contrast to positive "bioprecipitation" feedback caused by IN active bioparticles (Morris et al., 2014), the CCN-active SPP suppress precipitation. Model calculations indicate that when the number of SPP is equal to $10^6$, the suppression effect in clean conditions is 32%, while their lower value ($10^3$)

produces a negligible effect on precipitation.





Almost all angiosperm pollen grains are covered by a viscous material called *pollenkitt*, comprising saturated and unsaturated lipids, carotenoids, flavonoids, proteins, and carbohydrates (Piffanelli et al., 1998; Pacini and Hesse, 2005; Chichiriccò et al., 2019). The CCN properties of the submicron pollenkitt particles obtained from six different plant species were studied by Prisle et al.

(2019). It was shown that some pollenkitt species are surface-active and therefore decrease surface tension. The CCN-derived $\kappa$ values were generally between 0.1 and 0.2. For all species studied, the maximum $\kappa$ was observed at activation diameters ranging between 50-70 nm and steeply decreased above ~70 nm. Prisle et al. (2019) suggested that this behavior reflects the impact of pollenkitt surface activity on CCN activation, which decreases with particle growth factor increasing.

To our knowledge, no hygroscopic properties of SPP in subsaturated conditions have been presented in literature. Similar to other atmospheric aerosol particles, the hygroscopicity of SPP influences their life cycle in the atmosphere and the related direct and indirect effects on radiation budget. In addition, SPP hygroscopicity is a key factor enhancing the allergenic potential of chemically modified pollen allergens, like proteins (Pöschl et al., 2015; Reinmuth-Selzle et al., 2017). It was

documented that chemical modification of proteins by atmospheric nitrogen dioxide ($NO_2$) and ozone ($O_3$) proceeds efficiently at high RH within the particle bulk. The rate of the protein nitration and dimerization in the liquid phase occurred almost one order of magnitude higher in comparison to amorphous and semi-solid SPP states (Liu et al., 2017).

In this study, we investigated the hygroscopic properties and CCN activity of birch, pine and

rapeseed SPP for water saturation ratios between 0.02 and 1.012. These investigations were complemented with different chemical analyses of the subpollen components. A separate HHTDMA experiment was performed to evaluate the size-dependent restructuring of aerosol particles to correct their hygroscopic growth and CCN activity. We also conducted gravimetrical measurements coupled with dynamic light scattering, which allowed estimating the ffect of the colloid nanoparticles on the

SPP hygroscopicity. Together with the experimental analysis the κ-Köhler model was applied to reconcile the hygroscopicity and cloud condensation activity of subpollen particles.

## 2  Material and methods

### 2.1  Sample preparation

Pollen samples were collected from common genera belonging to the deciduous, coniferous and herbaceous plants. These species are: birch pollen (*Betula pendula*), pine pollen (*Pinus silvestris*), and rapeseed (*Brassica napus*), respectively. The aqueous pollen extracts were prepared by placing ~200 mg of pollen grains in 100 ml MilliQ-water (2mg ml$^{-1}$). The solutions were then 40 min extracted in an ultrasonic bath (frequency, 35 kHz; Bandelin, Sonorex Super 10P, Germany) in 1 min on /off

intervals. The aqueous extracts were filtered through a 0.45 µm syringe filter device (25mm GD/X,





sterile, 6901-2504, GE Healthcare Life Science, Whatman). The resulting filtered solution was atomized and dried to generate particles for HHTDMA and CCN measurements. The same extraction protocol was used for chemical and gravimetric analysis. It is expected that the aqueous extracts contain both surface substances (Suppl. S1) and cytoplasmic material released either due to osmotic shock or/and pollen rupture (Pummer et al., 2012; Steiner et al., 2015). As shown by Laurence (2011) the pollen degradation occurs more efficiently under sonification, therefore we used an ultrasonic bath.

## 2.2 Gravimetric analysis

Hereinafter we consider that the total solid (TS) in the filtered solution is the sum of both, the total dissolved species and total suspended solids (TSS) (hydrosols). The TS mass of subpollen species in the filtered solution was determined gravimetrically as follows. Aqueous pollen extracts were prepared as described in Sect. 2.1 with the exception that 100 mg of pollen grains were suspended in 50 ml MilliQ-water (2 mg ml$^{-1}$). After filtration the loaded syringe filter (25 mm, 0.45 µm) was first kept in a desiccator to gently remove liquid water and then dried in vacuum for 24 h (residual pressure 1 hPa). The weighing for the gravimetric PM determination was done with a microbalance (0.01 mg sensitivity) at 25°C and ~ 30 % RH, and the filters and impaction substrates were equilibrated at these conditions for 24 h prior to weighing. The TS was then determined as the difference between the initial pollen mass and the mass captured by the syringe filter (Suppl.S2, Table S.1). The mass ratio (*MR*) of the penetrated material through the filter to initial mass was further used to calculate the mass fraction ($w_i$) of chemical species: $w_i = C_i/MRC_0$, where $C_i$ is the mass concentration of species (*i*) in the filtered solution and $C_0$ is the initial mass concentration of pollen grains in water, respectively. To determine $C_i$ of the water-soluble species chemical analyses were performed as described below. The total mass concentration of water-insoluble species was determined as $C_{TSS} = C_{TS} - \sum_i C_i$ .

## 2.3 Chemical analysis

### 2.3.1 Materials

The following chemical and materials were used in this study: Ultra pure water from a GenPur™ UV-TOC/UFxCAD Plus water purification system (Thermo Scientific, Braunschweig, Germany), D-(+)-Glucose (> 99,5%, catalog #G8270, Sigma Aldrich, USA), phenol (> 99%, catalog 33517, Sigma Aldrich, USA), sulfiric acid (98%, catalog #1120801000, Merck, Darmstadt, Germany), Qubit® Assay Tubes (clear 0.5ml PCR tubes, Q32856, Thermo Scientific, Braunschweig, Germany), Qubit™ Protein assay kit (Q33211, 0.25-5µg, Invitrogen, Thermo Fisher Scientific, Braunscheig, Germany), Qubit™ dsDNA HS assay kit (Q32854, 0.2-100ng, Invitrogen, Thermo Fisher Scientific, Braunschweig, Germany), protein LoBind tubes (100 safe-lock tubes, PCR clean, 0030108132, Eppendorf, Hamburg, Germany), starch assay kit (SA20, Sigma Aldrich, USA).



### 2.3.2 Carbohydrate analysis

According to the protocol by Masuko et al. (2005) D-(+)-Glucose standard solutions were prepared in pure water at different concentrations (50, 250, 500, 1000, 2000, 3000 µM). 2.5 % (w/v) phenol solution was prepared (0.25 g phenol in 5 ml pure water. 50 µl of the standard solution and of each sample were added to the 96 well-plate in triplicates. Immediately 30 µl of the phenol solution were added and the well plate was incubated for 5 min at 90 °C in a drying oven (Binder, Tuttlingen, Germany) of 490 nm using a microplate reader (Multiscan GO, Thermo Scientific, Braunschweig, Germany).

### 2.3.3 Starch analysis

**Water-extractable starch**

The assay was done according to the manufacture´s protocol (Starch Assy Kit, Sigma Aldrich). Briefly, 1 ml of each of the TS was added to 1 ml of the starch assay reagent containing 50 units/ml of amyloglucosidase, which is an enzyme catalyzing the hydrolysis of starch to glucose. In the second step, the mixture is added to the glucose assay reagent containing 1.5 mM nicotinamide adenine dinucleotide (NAD), 1.0 mM adenosine triphophosphate (ATP), 1.0 units/ml of hexokinase, and 1.0 unit/ml of glucose-6-phosphate dehydrogenase (G6PDH). The reaction of the assay involve the following steps:

$$\text{Starch} + (n-1)\ \text{H2O} \xrightarrow{\text{Amyloglucosidase}} (n)\ \text{Glucose}$$

$$\text{Glucose} + \text{ATP} \xrightarrow{\text{Hexokinase}} \text{Glucose-6-Phosphate} + \text{ADP}$$

$$\text{G6P} + \text{NAD} \xrightarrow{\text{G6PDH}} \text{NADH} + \text{6-Phosphogluconate}$$

The increase in NADH (reduced form of NAD) and thus in the absorbance at 340 nm is directly proportional to the final glucose concentration.

**Total starch (Resistant starch, extraction with DMSO)**

The solid pollen samples were prepared as followed: 50µg of rapeseed, birch and pine pollen were dissolved in 10 ml dimethyl sulfoxide (DMSO) (1742589, Fisher scientific) and 2.5ml 8M hydrochloric acid and were incubated for 30 min at 60° C in an oven. After cooling down to room temperature, 25 ml of pure water was added and the pH was adjusted to approximately 5 with 5N sodium hydroxide solution. The samples were then treated as the water-extracted samples described above and the starch content was determined by the starch assay kit.

### 2.3.4 Protein analysis


The protein concentration of the birch, pine and rapeseed pollen extracts was determined using a Qubit® protein assay Kit according to the manufacturer's instruction (Thermo Fisher Scientific, Braunschweig, Germany). Briefly, 10µl of standard and samples are diluted in Qubit®Buffer to a final assay volume of 200µl. After vortexing for 2-3 seconds and incubation for 15 min at room temperature,

the tubes were inserted in the Qubit® 3.0 fluorometer (Invitrogen, Thermo Fisher Scientific, Braunschweig Germany), and the samples and standards were measured in triplicates.

### 2.3.5 DNA analysis

The DNA concentration of aqueous pollen extracts was determined according to the manufacturer's instruction of the Qubit® dsDNA HS assay kit (Q32854, 0.2-100ng, Invitrogen, Thermo Fisher

Scientific, Braunschweig, Germany).

### 2.3.6 Ion analysis.

Capillary electrophoresis system "CAPEL-105/105M" (Lumex,) is equipped with variable UV-spectrophotometric detector, with a wavelength range of 190–400 nm. The specialized software "Elforan" (Lumex, Russia) was used for instrument control and data acquiring. Fused-silica capillaries

with external polyimide coating (Polymicro Technologies, Phoenix, AZ, USA), 50 µm, I.D., 360 µm, O.D., effective length of 50 cm and total length of 60 cm. Hydrodynamic sample injection: 1500 mbar×sec (30 mbar×50 sec).

### 2.4 Size-resolved cloud condensation nuclei (CCN) measurements

A detailed description of the operation of the CCN counter (CCNC) and the subsequent data analysis

can be found in Pöhlker et al. (2016), which is the basis for the CCN part of this study. Briefly, size-resolved CCN measurements were conducted using a continuous-flow stream wise thermal-gradient CCN counter (model CCN-200, DMT, USA) coupled to a Differential Mobility Analyzer (DMA) (model 3081, TSI, DMA) and a condensation particle counter (CPC) (model 3772, TSI, USA). The CCNC was operated at a total flow rate of 0.5 l min$^{-1}$ with a sheath-to-aerosol flow ratio of 10. The

water vapor supersaturation ($S$) was regulated by the temperature difference of the CCNC flow column ($\Delta$T) and calibrated using ammonium sulfate aerosol and activity parameterization Köhler model (AP3) as described in Rose et al. (2008). Based on the calibration procedure, the overall uncertainty of $S$ is estimated to be ~7 %. For each CCN measurement cycle, $S$ set to 10 different values ranging from 0.18 % to 1.24 %. At each diameter selected by the DMA ($D_b$), the number concentration of total

aerosol particles (condensation nuclei, CN), $N_{CN}$, was measured with the CPC, and the number concentration of CCN, $N_{CCN}$, was measured with the CCNC. The measured CCN activated fractions ($F_{N_{CCN}/N_{CN}}(D_b, S) = N_{CCN}(D_b)/N_{CN}(D_b)$) were corrected for multiply charged particles and fitted with a cumulative Gaussian distribution function as described in Pöhlker et al (2016):


$$F_{\frac{N_{CCN}}{N_{CN}}}(D_b, S) = a \left( 1 + erf \left( \frac{D_b - D_{b,a}}{\sigma_a \sqrt{2}} \right) \right), \tag{1}$$

where $erf$ is the error function, $a$ is half the maximum value of $F_{N_{CCN}/N_{CN}}$, $D_{b,a}$ is the dry particle diameter at $F_{N_{CCN}/N_{CN}} = a$, and $\sigma_a$ is the standard deviation of the cumulative distribution function (CDF). The CCN activation curves were also corrected for differences in counting efficiencies of the CCNC and the CPC (Rose et al., 2010). The following best-fit parameters were determined for each

supersaturation: the maximum activated fraction $MAF_F = 2a$, the midpoint activation diameter, $D_{b,a}$, and the CDF standard deviation, $\sigma_a$. $MAF_F$ typically equals unity, except for external mixture of CCN-active with CCN-inactive particles, whereby the difference in CCN activity is due to chemical composition and hygroscopicity. While $\sigma_a$ serves as indicator for the extent of external mixing and heterogeneity of particle's composition. Calibration aerosols composed of high-purity

ammonium sulfate exhibit small non-zero $\sigma_a$ values that correspond to ~3 % of $D_{b,a}$ and can be attributed to heterogeneities of the water vapor supersaturation profile in the CCNC or other non-idealities such as DMA transfer function and particle shape effects (Rose et al., 2008). Thus, normalized CDF standard deviations or "heterogeneity parameter" values of $\sigma_a/D_{b,a}$ ~ 3 % indicate internally mixed CCN, whereas higher values indicate external mixtures of particles with varying

chemical composition and hygroscopicity (Rose et al., 2008; 2010).

## 2.5  HHTDMA setup and modes of operation

The hygroscopic properties of the subpollen particles were measured in the 2-99.5 % RH range with a high humidity tandem differential mobility analyser (HHTDMA) described elsewhere (Mikhailov

and Vlasenko, 2020). Briefly, two separate atomizers (model 3076, TSI, USA) were used for aerosol particles generation: first of them to study aerosol hygroscopic properties, and second for RH determination using ammonium sulfate particles growth factors (Fig.1). The generated aerosol solution droplets were dried to a relative humidity of ~3 % in a Nafion MD-700 dryer (L=60 cm; residence time, RT = 27 s) and then in a silica gel diffusion dryer (SDD, RT= 63 s) to the residual

relative humidity of ~2% RH. The dry aerosol was passed through a bipolar charger ($^{85}$Kr) and a differential mobility analyzer (DMA1) (model 3081, TSI, USA) to select monodisperse particles. This near monodisperse aerosol was then humidified, and the resulting particle size after humidification was measured with the scanning mobility particle sizer (DMA2) (SMPS, model 3080, CPC 3772, AIM version 9.0.0.0, Nov. 11, 2010, all TSI, USA). The particle size distributions measured were fitted

with a log-normal distribution function (Origin 9 software), and the modal diameter ($D_b$) of the fit function was used for further data analysis. The residence time between the aerosol preconditioning system and DMA2 depends on the humidification mode; its minimum value is 6.5 s, which




corresponds to RT in the hydration operation mode (Fig.1). Both DMAs are thermally insulated and were operated with a closed loop sheath air setup.

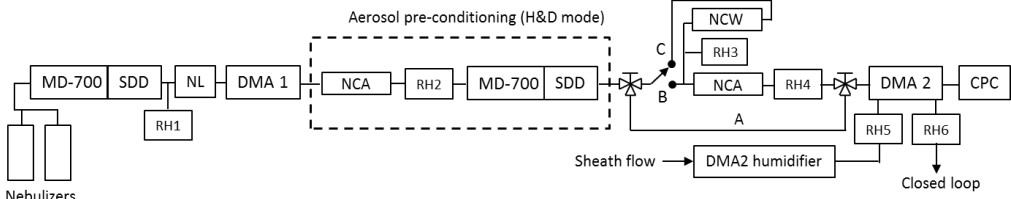

**Figure 1.** Experimental setup of the high humidity tandem differential mobility analyzer (HHTDMA) system: MD-700 – NAFION dryer, SDD – silica gel diffusion dryer, RH – relative humidity sensor, NL – 85Kr aerosol neutralizer, DMA – differential mobility analyzer, NCA – Nafion conditioner with air, NCW – Nafion conditioner with water, CPC – condensation particle counter, Operation mode: A– hydration&dehydration (H&D), B – hydration, C – dehydration.

**Table 1**. Sequence of relative humidity ("RH history") experienced by the investigated aerosol particles in the key elements of the HHTDMA system (DMA1, conditioner, DMA2) during different types of HHTDMA experiments (modes of operation). For each type of experiment, $X$ represents the independent variable, i.e., the RH value taken for plotting and further analysis of the measurement results.

| HHTDMA experiment (operation mode) | DMA2 (size selection) | Conditioner (humidification) | | DMA2 (size measurement) | |
|---|---|---|---|---|---|
| | RH1 (%) | RH2 (%) | RH3 (%) | RH4 (%) | RH5 (%) | RH6 (%) |
| Hydration and dehydration (H&D) | < 2 | $X$ | NU[a] | NU | < 2 | < 2 |
| Hydration | < 2 | RH$_{H\&D.min}$[b] | NU | $X$ | $X$ | $X$ |
| Dehydration | <2 | RH$_{H\&D.min}$ | > 96 | $X$ | $X$ | $X$ |

[a]NU − not used

[b]RH$_{H\&D,min}$ is the relative humidity that corresponds to the $D_{b.H\&,Dmin}$ obtained in H&D experiment.

5    The sheath and aerosol flow rates in both DMAs were 3.0 and 0.3 l min$^{-1}$, respectively. To control the RH we used a dew point probe (Dew Master, Edgetech Instrument, remote D-probe SC) and capacitive sensors (Almemo, FHAD 46C41A) in the range of 2-80 % RH and the ammonium sulfate scans at RH above 80 %. Based on the Extended Aerosol Inorganics Model (E-AIM, model II) (Clegg et al., 1998; Wexler and Clegg, 2002), we converted the measured ammonium sulfate growth factors into

10    RH($g_{b,E-AIM}$). The algorithm used for calculating the uncertainty of RH and growth factors is discussed in detail elsewhere (Mikhailov and Vlasenko, 2020). Figure 2 illustrates these uncertainties for the case of subpollen particles. One can see that throughout the all relative humidity range, the absolute RH uncertainty is less than 0.5 %, and the relative growth factor uncertainty due to RH and instrumental errors does not exceed 1 %.





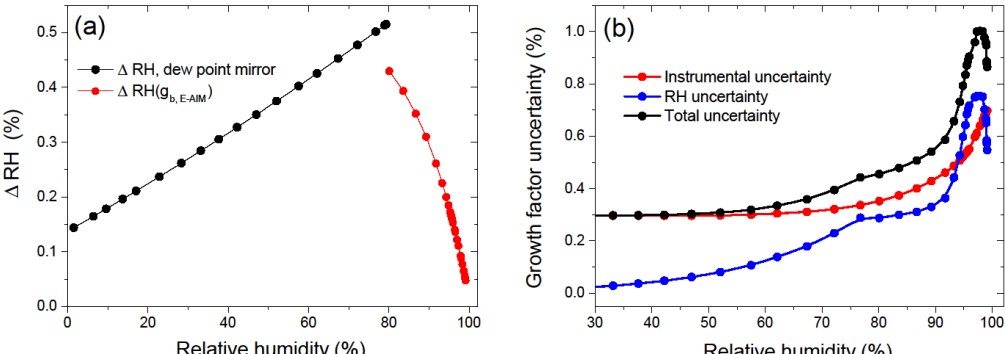

**Figure 2**. Accuracy in RH using different methods (**a**) and relative growth factor uncertainty due to instrumental and RH errors.

Three operation modes are available using this HHTDMA instrument: (A) hydration and dehydration (H&D) also called restructuring mode, (B) hydration, and (C) dehydration (Fig. 1). These correspond to different sequences of humidification and drying ("RH histories") of aerosol particles, as outlined in Table 1. The H&D mode was used to specify the optimal RH range in which initial irregular particles transform into compact globules (Fig.1, bypass, A). Upon hygroscopic growth study, the H&D mode was *in-situ* coupled with a conventional hydration or dehydration mode (Fig.1). The mobility equivalent particle growth factor, $g_b$ was calculated as the ratio of the mobility equivalent diameter, $D_b$ measured after conditioning (hydration, dehydration) to the minimum mobility diameter $D_{b,H\&D,min}$ observed in the H&D mode:

$$g_b = \frac{D_b}{D_{b,H\&D,min}} \quad . \tag{2}$$

### 2.6 Hydrodynamic size distribution of colloidal bioparticles

Aqueous extracts of pollen grains used for aerosol particle generation contain both, water-soluble and water-insoluble species. The presence of colloids in the size-selected particles can affect their hygroscopic properties. To evaluate this effect, we measured the hydrodynamic size distribution of colloidal bioparticles using a dynamic light scattering (DLS) system (model SZ-100, Horiba, Ltd). Measurement parameters were as follows: a laser wavelength of 532 nm (10 mW) a scattering angle of 90°, a measurement temperature of 25 °C, a medium viscosity of 0.896 mPa·s, a medium refractive index of 1.333, and material refractive index of 1.400. The size distribution of nano-bioparticles was determined in a filtered solution (1 ml) with the same concentration of pollen species as used for dry particle generation (Sect. 2.1). The samples were loaded into quartz microcuvettes, and 7-10 measurements were performed, for which average and standard deviation were calculated. The size





distribution obtained by DLS is based on the scattering intensity of the particles. For the case of Rayleigh particles the scattering intensity is proportional to the sixth power of the diameter. Thus, in term of intensity of light scattering the relative contribution, $f_{i,scat.}$ from each particle size bin is (Finsy, 1994; Li et al., 2014)

$$f_{i,scat.} = \frac{N_i D_i^6}{\sum N_i D_i^6} \qquad , \tag{3}$$

where $N_i$ is the number of particles in size bin $i$ having the mid-point diameter $D_i$. To convert the intensity-based size distribution into number particles size distribution ($f_{i,N}$) we let

$$x_i = \frac{f_{i,scat}}{D_i^6} \qquad . \tag{4}$$

Combining Eq. (3) and Eq. (4) we obtain normalized number particles size distribution:

$$f_{i,N} = \frac{x_i}{\sum x_i} \qquad . \tag{5}$$

As an example Fig. S2 shows the result of converting intensity-based size distribution of birch pollen colloids to a number-weighted distribution. If the suspended particles all have density $\rho$ then their mass $M_0$ with respect to the total number $N_0$, is

$$M_0 = N_0 \rho \frac{\pi}{6} \sum_i f_{i,N} D_i^3 \tag{6}$$

.

Total mass of suspended species in the filtered solution ($M_0$) was determined as the difference between the total mass of the solids in the filtered solution and the mass of dissolved species (Table 1), therefore $N_0$ can be calculated from Eq. (6). If $N_0$ and $f_{i,N}$ are known the number of colloids in each size bin is $N_i = f_{i,N} N_0$. A material density of 1.4 g cm$^{-3}$ was used for suspended organic particles suggesting that starch (1.53 g cm$^{-3}$), membrane proteins (1.37 g cm$^{-3}$), cellulose (1.5 g cm$^{-3}$) , carotenoids (~1 g cm$^{-3}$) and lipids (1.0 -1.2 g cm$^{-3}$ ) (Haynes, 2011) are the main species in the series of water-insoluble pollen compounds (Stanley and Linskens, 1974). In Eq. (6) $D_i$ is the hydrodynamic diameter obtained from the Stokes-Einstein relationship (Fucks, 1988). Since organic hydrosols are usually solvated and have an irregular shape, their hydrodynamic diameter can exceed the mass equivalent diameter (Maguare et al., 2018).

It should be noted that the DLS-based size distributions in some cases has a high degree of uncertainty. In the DLS setup, the light scattered by fluctuations of the concentration of molecules, particles, or aggregates suspended in a tested solution is recorded. To determine the rates of decay of the intensity of the scattered light, the time correlation function, $g(\tau)$ of this intensity is analyzed.



Since the correlation function is a superposition of exponential decays with distributed decay rates, the distribution function is the inverse Laplace transform of the correlation function. This transformation is carried out numerically by the CONTIN algorithm (Provencher, 1982; Scotti et al., 2015). One of the limitations of the resolution comes from the extremely ill-conditioned nature of this

Laplace inversion. Practically very small differences in $g(\tau)$ within typical experimental accuracy may result in quite different particle size distributions after inversion (Finsy, 1994; Anderson et al., 2013; Maguare et al., 2018; Varenne et al., 2016). Both DLS data uncertainty and inversion algorithm together with approximations used in Eq. (6) will provide an error in size distribution.

### 2.7 Aerosol particles shape

Inorganic and organic aerosol particles as well as their mixtures restructure upon humidification below its deliquescence. Irregular envelope shape and porous structure can cause a discrepancy between the mobility equivalent and mass equivalent particle diameters that usually limit precision of mobility diameter-based HTDMA and CCNC experiments (Mikhailov et al., 2004, 2009, 2020; Biskos et al., 2006; Gysel et al. 2004; Rose et al., 2008). To account for restructuring we used the minimum mobility

particle diameter, $D_{b,H\&D,min}$ obtained in H&D HHTDMA mode as an approximation of mass equivalent diameter of the dry solute particle, $D_s$ (i.e. $D_s = D_{b,H\&D,min.}$). Based on H&D measurements the dynamic shape factor, $\chi$ of the dry initial particles can be estimated as following (DeCarlo et al., 2004):

$$\chi = \frac{D_{b,i} C\left(D_{b,H\&D,min}\right)}{D_{b,H\&D,min} C\left(D_{b,i}\right)}, \tag{7}$$

where $D_{b,i}$ is the initial mobility equivalent diameter selected by DMA1 and measured by DMA2,

$C(D_{b,H\&D,min})$ and $C(D_{b,i})$ are the Cunningham slip correction factors for the respective diameters $D_{b,H\&D,min}$ and $D_{b,i}$. $\chi$ can be split into a component $\beta$ which describes the shape of the particle envelope and a component $\delta$ which is related to the particle porosity and allows the calculation of the void fraction inside the particle envelope $f$ (Brockmann and Rader, 1990):

$$\chi = \beta\delta \frac{C\left(D_{h\&d,min}\right)}{C\left(D_{h\&d,min}\delta\right)} \tag{8}$$

$$f = (1 - \delta^{-3}) \tag{9}$$

### 2.8  κ-Köhler model and effective hygroscopicity parameters

Both HTDMA and CCN data were fitted by the Köhler model introduced by P&K (Petters and Kreidenweis, 2007) to obtain an effective hygroscopicity parameter, κ, which is a measure of the





amount of water bound per unit volume of dry solute. The saturation ratio, $s$, over an aqueous solution
droplet is expressed in both sub- and supersaturated regimes by the κ-Köhler model:

$$s = a_w \times Ke = \frac{D_{wet}^3 - D_s^3}{D_{wet}^3 - D_s^3(1-\kappa)} \times exp\left(\frac{4\sigma_w M_w}{\rho_w RT D_{wet}}\right), \tag{10}$$

where $a_w$ is the water activity, $Ke$ is the Kelvin term, $D_s$ is the mass equivalent diameter of the dry
solute particle, $D_{wet}$ is the diameter of the solution droplet, $\sigma_w$ is the solution surface tension for pure
water (72 mNm[-1]), which is used here to produce a self-consistent data set as suggested in *P&K*, $M_w$
is the molecular weight of water, $\rho_w$ is the density of water, $R$ is the universal gas constant, $T$ is the
temperature. The hygroscopicity, $\kappa_{b,a}$ from CCN data set was obtained by inserting the CCN-derived
dry activation diameter $D_{b,a}$ for $D_s$ and varying both $\kappa$ and $D_{wet}$ until the saturation ratio $s = RH/100$
% was equivalent at the same time to the prescribed supersaturation $S = (s-1)100$ % and to the
maximum of a Köhler model curve of CCN activation. The same procedure was used to calculate the
corrected hygroscopicity $\kappa_a$ due to irregularity shaped structure of the initial dry particles (Sect. 3.3;
Eq.17). In this case, the restructured diameter, $D_a$ was inserted for $D_s$. Alternatively, the
hygroscopicity parameter can be determined from an approximate formula (Petters and Kreidenweis,
2007):

$$\kappa_{app} = \frac{4A^3}{27 D_a^3 ln^2 s_c},$$

$$A = \frac{4\sigma M_w}{\rho_w RT}$$
$$. \tag{11}$$

For the studied samples the difference between $\kappa_a$ and $\kappa_{app}$ calculated from Eq.(10) and its simplified
form (Eq.11) for $\kappa > 0.07$ on average did not exceed 2%. Therefore, $\kappa_a$ uncertainty was calculated as
Gaussian propagated error based on Eq. (11) using experimental uncertainties associated with $s_c$ and
$D_a$:

$$\Delta\kappa_a = \sqrt{\left(-\frac{12A^3}{27 D_a^4 ln^2 s_c}\Delta D_a\right)^2 + \left(-\frac{8A^3}{27 s_a D_a^3 ln^3 s_c}\Delta s_a\right)^2} \tag{12}$$

Combining water activity, $a_w$ from Eq. (10) and Eq.(2) gives:

$$\kappa_b = \frac{(g_b^3 - 1)(1 - a_w)}{a_w}, \tag{13}$$





where $\kappa_b$ refers to the HHTDMA-measured hygroscopicity $\kappa$. The uncertainty for $\kappa_b$ was obtained as propagated error based on Eq. (13). To convert the measured RH-based growth curves ($g_b$ vs. $RH$) into activity-based curves ($g_b$ vs. $a_w$) we divided the RH values through the Kelvin term of Eq.(10) assuming that $\sigma$ is equal to surface tension of pure water (72 mN m$^{-1}$).

As proposed by Kreidenweis et al. (2005), hygroscopic growth curves can be approximately described with a polynomial three-parameter fit function of the following form:

$$g_b = \left(1 + [k_1 + k_2 a_w + k_3 a_w^2]\frac{a_w}{1 - a_w}\right)^{1/3}, \qquad (14)$$

where adjustable parameters $k_1$, $k_2$, and $k_3$ capture non-ideality, caused by mixing between water and solutes. From Eqs. (13) and (14) the dependence of $\kappa_b$ on $a_w$ can be described by:

$$\kappa_b = k_1 + k_2 a_w + k_3 a_w^2 \qquad (15)$$

## 3   Experimental results and discussion

### 3.1   Chemical composition of pollen species

The results of the chemical analysis of water-extractable compounds in the filtered solution are summarized in Table 2 and illustrated in Fig. 3. Their mass fraction, $w_i$ was determined as described in Sect. 2.2. According to the chemical and gravimetrical analysis, the fraction of the total dissolved species (TS-TSS) varies from ~35 wt % (pine and rapeseed samples) to ~ 65 wt % (birch sample).

Among them the major components are carbohydrates (i.e. ~28 wt % for birch and rapeseed and 51 wt % for pine pollen solution) followed by inorganic ions (< 10 wt %). The most common soluble carbohydrates in pollen grains are monosaccharides (glucose, fructose) and disaccaharydes (sucrose, maltose). Other soluble carbohydrates such as raffinose, stachyose, rhamnose, and arabinose can persist in a lesser amount (Stanley and Linskens, 1974). Water-extractable starch in the filtered

solution is 2, 1, and 0.8 wt % for birch, pine and rapeseed solution, respectively (Table 2). However, extractable starch contain only minor part of resistant starch measured in pollen grains as described in Sect. 2.3.3. The mass ratio of water-extractable to resistant starch is ~ 9 % (birch), ~ 3% (pine), and ~ 7% (rapeseed) by weight (not shown in Table 2). Protein is the second most abundant organic component after carbohydrates. Its content in the filtered extracts varies from ~ 0.3 wt % (pine pollen)

to ~ 3 wt % (birch pollen). DNA in negligible quantities was detected in birch and pine aqueous extracts. Phosphorus in the form of PO$_4^{3-}$ and K$^+$ are the dominant inorganic ions and account for ~ 3 wt % of the extractable material. All filtered pollen extracts contain a small amount of SO$_4^{2-}$ ion varying from 0.3 wt % (birch pollen extract) to 2 wt % (pine pollen extract).



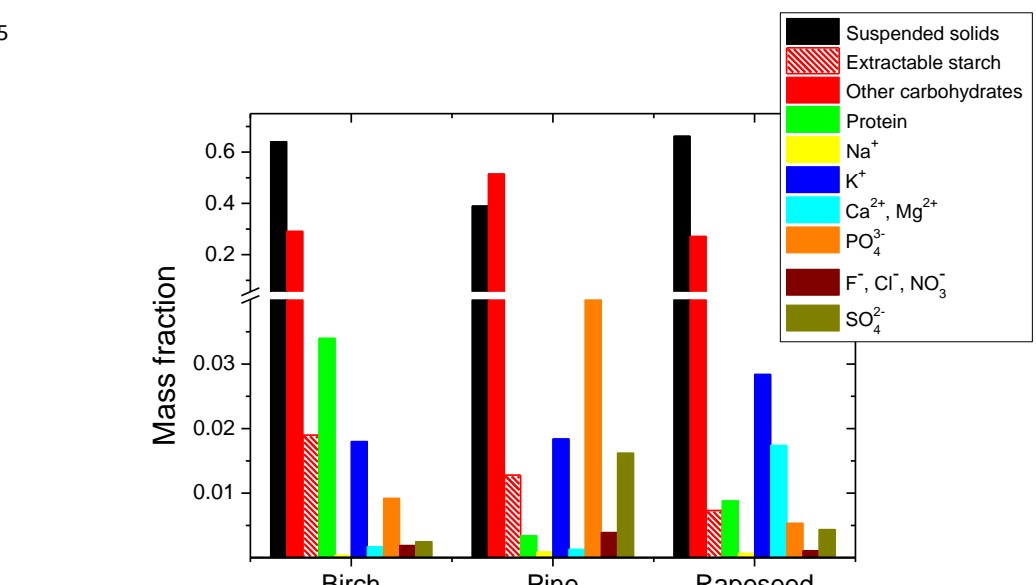

**Figure 3.** Mass fraction of dissolved and suspended species in the filtered solution of pollen grains.

Calcium and magnesium are also present in the filtered pollen solutions. Their highest total value of ~ 1 wt % was found in the rapeseed sample (Table 1). Other ions such as $Na^+$, $F^-$, $Cl^-$ and $NO_3^-$ were found in small amounts. Their total mass ratio does not exceed 0.3 wt %. The biological role of the above chemical species during pollen grains development and maturation is considered elsewhere

10 (Baker and Baker, 1979; Pacini, et al., 1999; Roulston et al., 2000; Stanley and Linskens, 1974). Based on the chemical analysis results, it is reasonable to assume that water-soluble sugars, proteins, and inorganic ions are the main components that causing water uptake by pollen subparticles. Table 2 shows that the mass ratio of water-insoluble compounds in the filtered solutions is significant with amounts up 39, 64 and 66 wt % of pine, birch and rapeseed extracts, respectively. These suspended

15 particles (hydrosols or colloids) can comprise cross-linked sporopollenin, callose, carotenoids, cellulose, starch, and sterols in various proportions (Pacini and Hesse, 2005; Stanley and Linskens, 1974). We consider this fraction of pollen species as an inert material that suppresses hygroscopic growth and CCN activity of subpollen particles, as discussed below.




**Table 2.** Concentration ($\mu$g ml$^{-1}$) and mass fraction ($w_i$) of chemical species ($i$) in the filtered solution of pollen grains. Total solids is the sum of the total dissolved species and total suspended solids. $w_i$ is the ratio of the amount of selected species to the mass of total solids containing in 1 mL of water.

| Chemical species | Birch | | Pine | | Rapeseed | |
|---|---|---|---|---|---|---|
| | $\mu$g ml$^{-1}$ | $w_i$ | $\mu$g ml$^{-1}$ | $w_i$ | $\mu$g ml$^{-1}$ | $w_i$ |
| Carbohydrates | 190 ± 8 | 0.29 | 299 ± 11 | 0.51 | 252 ±15 | 0.27 |
| Water- extractable starch | 12.4 ± 8.0 | 0.02 | 7.4 ± 5.9 | 0.01 | 6.8 ± 7.2 | 0.008 |
| Protein | 22.0 ± 3.8 | 0.03 | 2.00 ± 0.15 | 0.003 | 8.22 ± 0.19 | 0.009 |
| DNA | 0.018 ± 0.002 | $2.7 \times 10^{-5}$ | $1.6 \times 10^{-4}$ | $< 3 \times 10^{-7}$ | - | - |
| Na$^+$ | 0.275 ± 0.015 | $4.2 \times 10^{-4}$ | 0.52 ± 0.01 | $9.0 \times 10^{-4}$ | 0.62 ± 0.02 | $6.7 \times 10^{-4}$ |
| K$^+$ | 11.77 ± 0.03 | 0.02 | 10.69 ± 0.03 | 0.02 | 26.44 ± 0.64 | 0.03 |
| Ca$^{2+}$ | 0.65 ± 0.01 | 0.001 | 0.35 ± 0.01 | $6.0 \times 10^{-4}$ | 11.30 ± 0.23 | 0.01 |
| Mg$^{2+}$ | 0.46 ± 0.01 | 0.001 | 0.41 ± 0.01 | $7.1 \times 10^{-4}$ | 4.90 ± 0.04 | $5.2 \times 10^{-3}$ |
| F$^-$ | 0.46 ± 0.02 | 0.001 | 0.18 ± 0.01 | $3.1 \times 10^{-4}$ | 0.60 ±0.03 | $1.3 \times 10^{-4}$ |
| Cl$^-$ | 0.78 ± 0.06 | 0.001 | 1.19 ± 0.08 | $2.1 \times 10^{-3}$ | 0.89 ±0.01 | $9.6 \times 10^{-4}$ |
| NO$_3^-$ | - | - | 0.90 ± 0.02 | $1.5 \times 10^{-3}$ | - | - |
| PO$_4^{3-}$ | 5.97 ± 0.17 | 0.009 | 29.7 ± 0.9 | 0.05 | 4.96 ± 0.08 | 0.005 |
| SO$_4^{2-}$ | 1.64 ± 0.01 | 0.003 | 9.42 ± 0.14 | 0.02 | 4.06 ± 0.08 | 0.004 |
| All ions | 22.0 ± 0.3 | 0.04 ± 0.01 | 53.3 ± 1.2 | 0.09 ± 0.01 | 53.3 ± 1.1 | 0.06 ± 0.01 |
| All dissolved organics | 212 ± 21 | 0.32 ± 0.03 | 301 ± 17 | 0.52 ± 0.03 | 261 ± 16 | 0.28 ± 0.02 |
| Total dissolved species | 234 ± 13 | 0.36 ± 0.04 | 354 ± 12 | 0.61 ± 0.03 | 313 ± 17 | 0.34 ± 0.02 |
| Total suspended solids | 418 ± 17 | 0.64 ± 0.03 | 227 ± 20 | 0.39 ±0.05 | 616 ±19 | 0.66 ± 0.03 |
| Total solids | 652 ± 4 | 1.00 ±0.05 | 581 ± 16 | 1.00 ±0.08 | 930 ± 9 | 1.00 ± 0.02 |

## 3.2 Hydrodynamic size distribution of hydrosols

In addition to the mass fraction of suspended particles in the nebulized solution, their effect on the hygroscopic and CCN properties depends also on their size. Figure 4 shows the hydrodynamic size distribution of colloidal bioparticles determined by a DLS system using the conversion algorithm described in Sect. 2.6. As expected, all size distributions are highly uncertain, especially in case of aqueous suspensions of birch (Fig.4a) and pine (Fig.4b) samples. Nevertheless, the obtained results clearly show that the size of colloidal particles spans the range of 40-110 nm, which is within the SPP size range of dry particles (20-190 nm) used for CCN and HHTDMA measurements. It follows that the studied subpollen particles are a mixture of water-soluble and water-insoluble (suspended) compounds in different proportions (Table 2). Besides, colloidal particles tend to coagulate, forming aggregates. This aging process can lead to a different composition of the dry particles during the HHTDMA or CCN experiments. A simple optical experiment with an aqueous extract of birch pollen





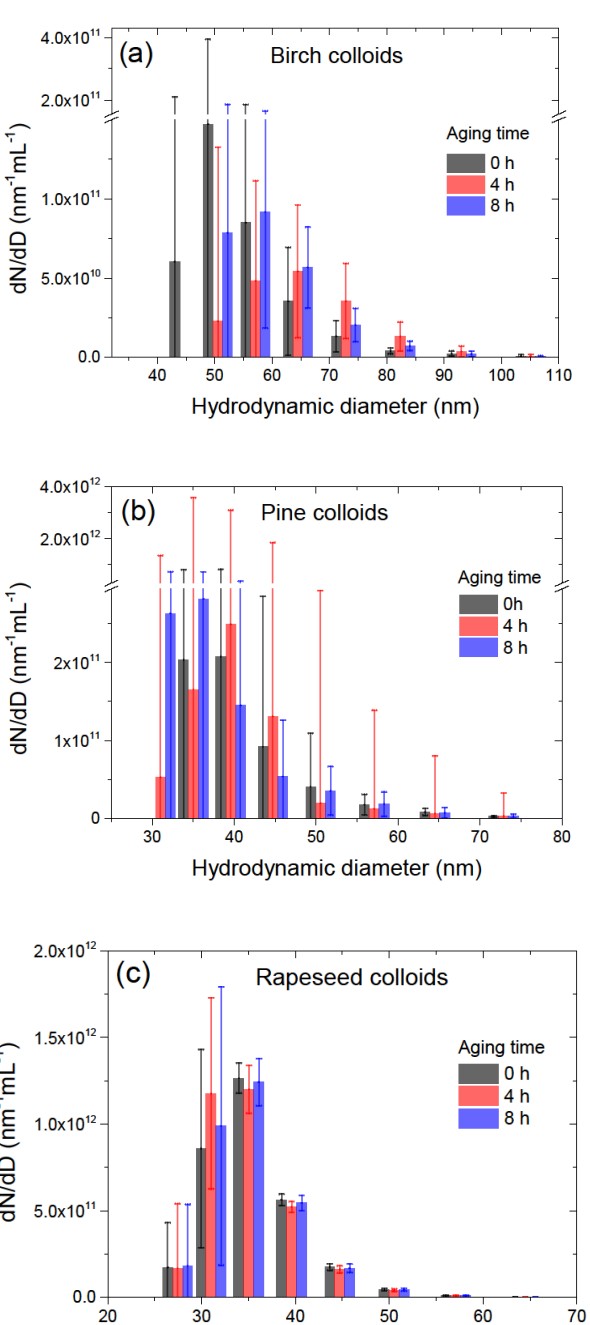

**Figure 4.** Colloidal particles number size distribution of aqueous extracts of birch (**a**), pine (**b**) and rapeseed (**d**) pollen grains**:** initial, (0 h)  and after 4h and 8 h aging, respectively.





showed that coagulation of biopolymers causes light scattering on colloids (Tyndall effect). At the used concentrations, however, this effect becomes noticeable only after 10 hours aging (Suppl. S5, Fig. S.3). Figure 4 shows that for a typical time scale of CCN (~ 4 h) and HHTDMA (~ 8 h) experiments, the size distributions of birch, pine and rapeseed colloids remain constant within the

uncertainty range, indicating that coagulation of suspended particle is small and does not lead to a noticeable change in aerosol composition in the specified time periods.

## 3.3 Particles restructuring

The H&D HHTDMA operation mode was first used to study the size-dependent aerosol particle restructuring ($D_{b,i}$ = 30–180 nm). The dried aerosol particles selected by DMA1 entered to the pre-

conditioning section (Fig.1, dashed rectangle), underwent microstructural transformation, acquiring a more compact/spherical microstructure, during a cycle of humidification (H1, RT = 0.5 s) and drying (Nafion MD-700, RT = 27 s; SDD, RT = 16 s) (Mikhailov et al., 2004; 2009). The particles growth factor obtained during the H&D experiment was calculated as follows: $g_{H\&D} = D_{b,RH}/D_{b,i}$, where $D_{b,RH}$ is the mobility particle diameter measured at RH = RH2 (Table.1).

15         Figure 5 illustrates the measurement of the subpollen particles growth factors in H&D mode. All species exhibit restructuring, i.e. the growth factors decrease with increasing RH, wherein this effect is strongest for largest diameters. It is also seen that for birch and pine SPP the $g_{H\&D,min}$ is already reached at ~ 35 % RH, while for rapeseed SPP the $g_{H\&D,min}$ values are observed at higher humidity starting from ~ 65 % RH (excluding particles with $D_{b,i} < 60$ nm). The size-dependent

$RH_{H\&D,min}$ intervals used to $g_{H\&D,min}$ averaging are shown in Table 3. The $RH_{H\&D,min}$ range for ~100 nm particles was further employed in HHTDMA hydration/dehydration experiments to convert *in-situ* irregular initial particles into compact globules. Figure 6 shows the growth factors, $g_{H\&D,min}$ obtained over the $D_{b,i}$ range of 30–180 nm together with an exponential fitting curve,

$$g_{b,H\&D,min} = \eta + \varphi exp\left(-\frac{D_{b,i}}{\tau}\right), \tag{16}$$

where η, φ, and τ are the best fit parameters listed in Table S 2. The fit values of $g_{H\&D,min}$ are

further used for particles shape correction of initial dry activation diameters, $D_{b,a}$ determined in size-resolved CCN experiments:

$$D_a = g_{b,H\&D,min}D_{b,a} . \tag{17}$$

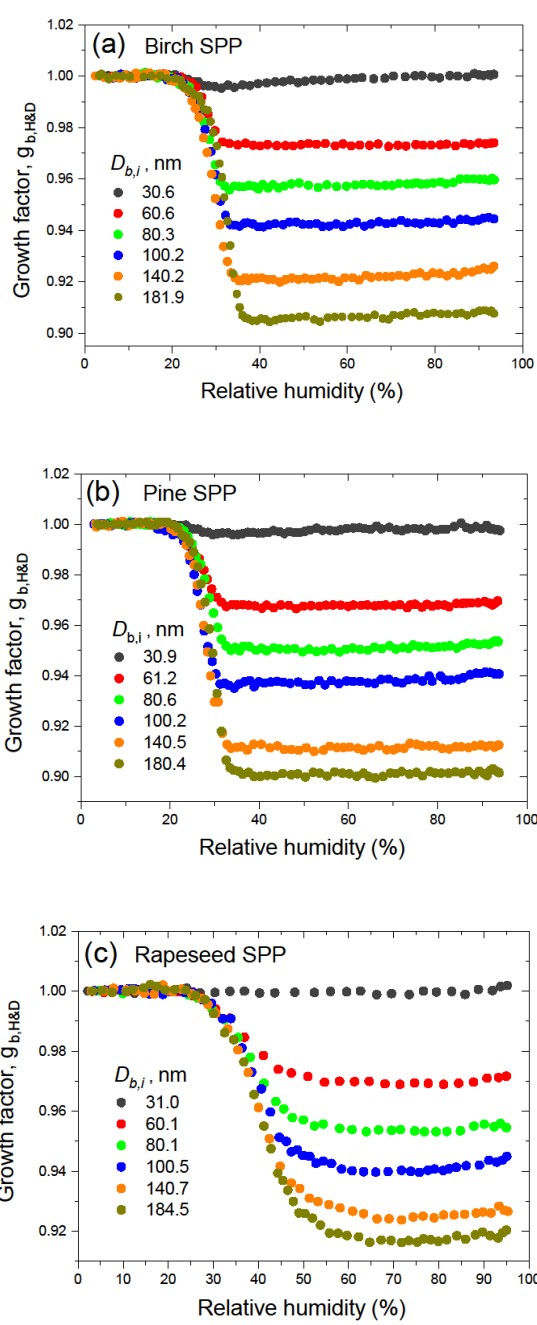

**Figure 5.** Growth factors ($g_{b,H\&D}$) of birch **(a)**, pine **(b)** and rapeseed **(c)** SPP as a function of relative humidity observed for different initial mobility diameters ($D_{b,i}$) in hydration&dehydration (H&D) HHTDMA experiments.





**Table 3.** The size-dependent microstructural rearrangement parameters of subpollen particles (SPP) obtained in H&D experiment. $D_{b,i}$ and $g_{b,H\&D,min}$ are mean (± 2 SD ) initial mobility diameter and minimum mobility growth factor, respectively. The dynamic shape factor ($\chi$), porosity ($\delta$), and void fraction ($f$) together with propagated error ($\Delta f$) are calculated from Eq.(7), Eq.(8), and Eq.(9), respectively. For birch, pine, and ~30 nm rapeseed SPP the shape particle envelope ($\beta$) is set to 1; for rapeseed SPP at $D_{b,i} > 30$ nm the $\beta$ values were obtained as described in Suppl. S4). $RH_{b,H\&D,min}$ is the relative humidity at which the minimum mobility growth factor was observed.

| $D_{b,i}$ (nm) | $g_{b,H\&D,min}$ | $\chi$ | $\beta$ | $\delta$ | $f \pm \Delta f$ (%) | $RH_{H\&D,min}$ (%) |
|---|---|---|---|---|---|---|
| Birch SPP | | | | | | |
| 30.56 ± 0.04 | 0.997 ± 0.002 | 1.006 | 1 | 1.003 | 0.9 ± 1.0 | 28 – 36 |
| 60.59 ± 0.05 | 0.973 ± 0.001 | 1.051 | 1 | 1.028 | 7.9 ± 0.4 | 33 – 50 |
| 80.32 ± 0.12 | 0.957 ± 0.002 | 1.082 | 1 | 1.045 | 12.4 ± 0.7 | 33 – 50 |
| 100.21 ± 0.08 | 0.942 ± 0.001 | 1.110 | 1 | 1.061 | 16.3 ± 0.5 | 33 – 50 |
| 140.18 ± 0.20 | 0.921 ± 0.002 | 1.146 | 1 | 1.085 | 21.7 ± 0.5 | 34 – 50 |
| 181.86 ± 0.08 | 0.906 ± 0.001 | 1.169 | 1 | 1.104 | 25.7 ± 0.5 | 38 – 54 |
| Pine SPP | | | | | | |
| 30.95 ± 0.02 | 0.996 ± 0.001 | 1.007 | 1 | 1.004 | 1.1 ± 0.4 | 29 – 40 |
| 61.19 ± 0.07 | 0.968 ± 0.003 | 1.062 | 1 | 1.033 | 9.3 ± 1.2 | 33 – 50 |
| 80.58 ± 0.06 | 0.950 ± 0.002 | 1.096 | 1 | 1.052 | 14.1 ± 0.6 | 35 – 58 |
| 100.20 ± 0.15 | 0.937 ± 0.002 | 1.121 | 1 | 1.067 | 17.7 ± 0.8 | 33 – 58 |
| 140.45 ± 0.17 | 0.911 ± 0.002 | 1.166 | 1 | 1.097 | 24.3 ± 0.7 | 33 – 60 |
| 180.41 ± 0.20 | 0.901 ± 0.002 | 1.179 | 1 | 1.110 | 26.9 ± 0.5 | 39 – 62 |
| Rapeseed SPP | | | | | | |
| 31.03 ± 0.04 | 0.999 ± 0.002 | 1.002 | 1 | 1.001 ± 0.003 | <1 | 65 – 85 |
| 60.06 ± 0.04 | 0.969 ± 0.001 | 1.060 | 1.03 ± 0.02 | 1.02 ± 0.02 | 4.9 ± 6.2 | 75 – 85 |
| 80.12 ± 0.02 | 0.953 ± 0.001 | 1.090 | 1.03 ± 0.02 | 1.03 ± 0.02 | 8.8 ± 4.3 | 61 – 82 |
| 100.50 ± 0.02 | 0.940 ± 0.001 | 1.114 | 1.04 ± 0.03 | 1.04 ± 0.02 | 11.1 ± 5.3 | 70 – 82 |
| 140.71 ± 0.15 | 0.925 ± 0.002 | 1.138 | 1.07 ± 0.03 | 1.04 ± 0.02 | 11.4 ± 6.2 | 66 – 80 |
| 184.47 ± 0.12 | 0.917 ± 0.002 | 1.146 | 1.09 ± 0.02 | 1.03 ± 0.02 | 9.3 ± 5.2 | 66 – 82 |

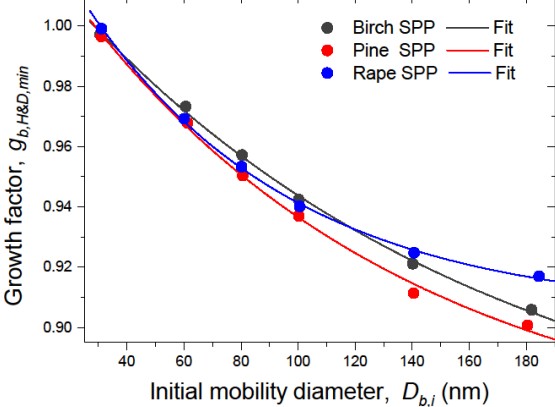

**Figure 6.** Minimum mobility growth factors ($g_{b,H\&dD,min}$) of birch, pine, and rapeseed subpollen particles as a function of initial mobility diameter ($D_{b,i}$). Line is the exponential fit function with the best fitting parameters listed in Table S2.





### 3.4 Dry particles morphology

Using Eq. (7) and assuming that $D_{b.H\&D,min}$ obtained in H&D mode is equal to $D_s$ we calculated the size-dependent dynamic shape factor of the SPP, $\chi$ (Table 3). Scanning electronic microscope (SEM) images of the initial birch and pine SPP show a spherical shape (Fig.7a and 7b), therefore, one can

accept that $\beta=1$. Using Eq. (8) and Eq. (9) we estimated the porosity ($\delta$) and void fraction ($f$) of birch and pine SPP depending on their size (Table 3).

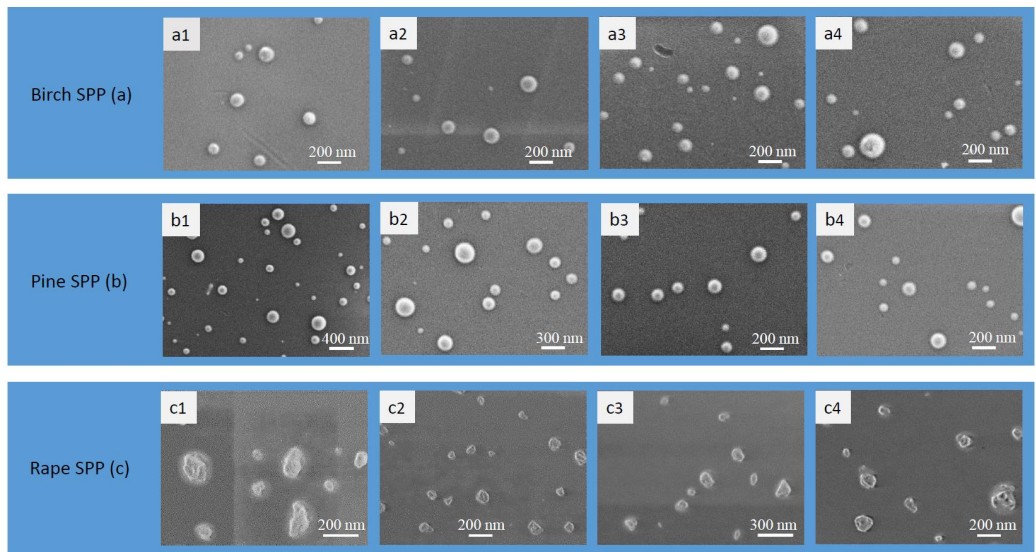

**Figure 7.** SEM images of birch (a), pine (b), and rape (c) subpollen particles. The samples were investigated with a high-resolution SEM (ZEISS Merlin). Operation conditions: 0.4 kV accelerating voltage, In-Lens detector of SE, 3.6 mm working distance. Particle samples were collected directly onto a 3mm TEM copper 300 mesh grids, coated with a 30–60 nm thick Formvar film.

The obtained results indicate that despite the spherical shape the initial dry particles have a porous structure, wherein the void fraction gradually increases with their size, reaching ~26 % for particles

of ~180 nm. Figure 7c shows that in contrast to birch and pine SPP the rapeseed particles external morphology is not spherical. To estimate the envelope component $\beta$ of rapeseed SPP we used Eq. (S1), approximating their shape by prolate ellipsoid as described in Suppl. S.4. The resulting microstructural parameters of rapeseed SPP calculated from Eq. (8) and Eq. (9) are listed in Table 3. Note that $\delta$ and $f$ values obtained in this way are their upper estimate, since it is assumed that all

particles are oriented in both DMA to its flow by maximal axis. In addition, due to technical limitation of SEM for 30 nm particles the $\beta$ was set to 1. Nevertheless, even under these assumptions, the void fraction in rapeseed particles is on average 2-fold less than that for birch and pine particles (Table 3). Note that according to chemical analyses (Table 2), rapeseed extracts contain the highest amount of





calcium (~11 µg ml$^{-1}$) and undissolved species (~620 µg ml$^{-1}$). Since calcium is mainly present in the walls of the pollen grains, it is reasonable to assume that the irregular morphology of rapeseed SPP (Fig. 7c) is due to the predominant aggregation of exine and intine species such as sporopollenin and cellulose (Stanley and Linskens, 1974).

### 3.5 CCN properties

The average parameters derived from CCN activation curves as a result of three repeated measurements for each sample are summarized in Table 4. Figure 8 shows the maximum activation
10    fraction (MAF$_F$) (panel **a**) and normalized standard deviations ($\sigma_a/D_{b,a}$) (panel **b**) against the corresponding midpoint activation diameters ($D_{b.a}$) and supersaturation (S %) obtained in a series of three CCN measurements for each type of SPP.

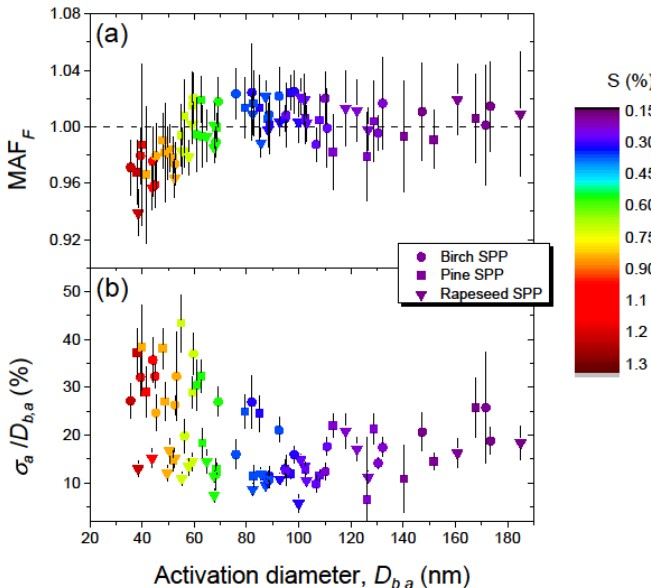

**Figure 8**. Characteristic parameters of birch, pine and rapeseed subpollen particles derived from CCN activation curves: maximum activated fraction (MAF$_F$) (**a**) and heterogeneity parameter ($\sigma_a/D_{b,a}$) (**b**).

Figure 8a shows that for $D_{b,a}$> 80 nm the $MAF_F$ is close to one and $\sigma_a/D_{b,a}$ (Fig. 8b) ranges from 10
15    % to 25 %, which implies that nearly all aerosol particles larger ~80 nm were CCN-active at $S$ smaller 0.5 %. As detailed in Sect. 2.4, $\sigma_a/D_{b,a}$ characterizes the heterogeneity of CCN-active particles in the size range around $D_{b,a}$. The observed 10-25 % heterogeneity range is clearly higher than the ~3% observed for aerosols of homogeneous chemical composition, indicating that the particles in this size



**Table 4.** Characteristic average CCN parameters (mean values ± SD) for subpollen particles (SPP) at different $S$ (%) obtained in a series of three measurements for each sample. Quantities are a maximum activated fraction, $MAF_F$, heterogeneity parameter, $\sigma_a/D_{b,a}$, initial, $D_{b,a}$, $\kappa_{b,a}$ and shape corrected, $D_a$, $\kappa_a$ dry diameter and hygroscopicity parameter, respectively. $\delta\kappa$ is the relative change in hygroscopicity due to particles restructuring, and $D_{wet,a}$ is the wet activation diameter.

| $S$ (%) | $MAF_F$ | $\sigma_a/D_{b,a}$ (%) | $D_{b,a}$ (nm) | $D_a$ (nm) | $D_{wet,a}$ (nm) | $\kappa_{b,a}$ | $\kappa_a$ | $\delta k$ (%) |
|---|---|---|---|---|---|---|---|---|
| **Birch SPP** | | | | | | | | |
| 0.18 ± 0.01 | 1.01 ± 0.04 | 21.8 ± 6.1 | 150 ± 4 | 137 ± 3 | 783 ± 44 | 0.11 ± 0.02 | 0.13 ± 0.02 | 18 ± 4 |
| 0.23 ± 0.02 | 1.01 ± 0.02 | 14.7 ± 1.6 | 125 ± 2 | 117 ± 2 | 659 ± 31 | 0.12 ± 0.02 | 0.15 ± 0.02 | 21 ± 4 |
| 0.28 ± 0.02 | 1.00 ± 0.02 | 13.4 ± 2.3 | 106 ± 2 | 100 ± 2 | 508 ± 18 | 0.15 ± 0.02 | 0.18 ± 0.03 | 17 ± 3 |
| 0.32 ± 0.02 | 1.02 ± 0.02 | 18.7 ± 3.0 | 95 ± 2 | 90 ± 2 | 433 ± 13 | 0.15 ± 0.03 | 0.18 ± 0.03 | 16 ± 3 |
| 0.37 ± 0.03 | 1.02 ± 0.02 | 16.0 ± 2.2 | 91 ± 2 | 86 ± 2 | 378 ± 10 | 0.13 ± 0.02 | 0.16 ± 0.02 | 13 ± 3 |
| 0.57 ± 0.05 | 1.00 ± 0.02 | 23.1 ± 3.9 | 79 ± 2 | 75 ± 1 | 251 ± 4 | 0.13 ± 0.02 | 0.15 ± 0.03 | 11 ± 2 |
| 0.71 ± 0.05 | 0.99 ± 0.02 | 27.2 ± 3.3 | 70 ± 1 | 67 ± 1 | 201 ± 3 | 0.14 ± 0.02 | 0.16 ± 0.02 | 10 ± 2 |
| 0.85 ± 0.06 | 0.98 ± 0.04 | 30.3 ± 5.1 | 60 ± 3 | 52 ± 3 | 168 ± 2 | 0.13 ± 0.03 | 0.14 ± 0.03 | 10 ± 2 |
| 1.04 ± 0.07 | 0.97 ± 0.02 | 31.8 ± 3.6 | 45 ± 1 | 44 ± 1 | 137 ± 2 | 0.14 ± 0.02 | 0.15 ± 0.03 | 11 ± 4 |
| All | | | | | | 0.13 ± 0.02 | 0.16 ± 0.03 | |
| **Pine SPP** | | | | | | | | |
| 0.18 ± 0.01 | 1.00 ± 0.03 | 17.0 ± 3.3 | 148 ± 16 | 134 ± 14 | 774 ± 43 | 0.13 ± 0.04 | 0.17 ± 0.05 | 23 ± 8 |
| 0.23 ± 0.02 | 0.99 ± 0.03 | 16.6 ± 2.2 | 124 ± 8 | 115 ± 7 | 616 ± 27 | 0.14 ± 0.03 | 0.17 ± 0.03 | 20 ± 4 |
| 0.28 ± 0.02 | 1.01 ± 0.02 | 12.3 ± 1.3 | 104 ± 4 | 97 ± 3 | 507 ± 18 | 0.16 ± 0.02 | 0.19 ± 0.02 | 18 ± 2 |
| 0.33 ± 0.02 | 1.01 ± 0.02 | 15.9 ± 2.3 | 91 ± 5 | 86 ± 5 | 434 ± 13 | 0.17 ± 0.03 | 0.20 ± 0.03 | 15 ± 3 |
| 0.37 ± 0.03 | 1.01 ± 0.02 | 15.9 ± 2.2 | 85 ± 5 | 81 ± 5 | 378 ± 10 | 0.16 ± 0.03 | 0.18 ± 0.03 | 14 ± 3 |
| 0.56 ± 0.04 | 1.00 ± 0.02 | 21.2 ± 2.5 | 73 ± 17 | 70 ± 15 | 252 ± 4 | 0.16 ± 0.02 | 0.17 ± 0.03 | 10 ± 6 |
| 0.71 ± 0.05 | 1.01 ± 0.02 | 30.7 ± 3.5 | 66 ± 18 | 64 ± 16 | 201 ± 3 | 0.15 ± 0.02 | 0.16 ± 0.02 | 8 ± 5 |
| 0.85 ± 0.06 | 0.98 ± 0.02 | 26.6 ± 3.1 | 56 ± 12 | 54 ± 11 | 167 ± 2 | 0.15 ± 0.02 | 0.16 ± 0.02 | 6 ± 4 |
| 1.04 ± 0.07 | 0.97 ± 0.05 | 29.0 ± 5.3 | 42 ± 2 | 41 ± 1 | 137 ± 2 | 0.17 ± 0.02 | 0.18 ± 0.02 | 6 ± 2 |
| 1.24 ± 0.09 | 0.98 ± 0.04 | 37.8 ± 7.0 | 39 ± 1.4 | 38 ± 1 | 116 ± 2 | 0.15 ± 0.02 | 0.16 ± 0.02 | 10 ± 4 |
| All | | | | | | 0.15 ± 0.04 | 0.18 ± 0.04 | |
| **Rapeseed SPP** | | | | | | | | |
| 0.18 ± 0.01 | 1.01 ± 0.03 | 17.3 ± 3.3 | 174 ± 19 | 159 ± 15 | 784 ± 43 | 0.11 ± 0.03 | 0.12 ± 0.03 | 8 ± 3 |
| 0.23 ± 0.02 | 1.01 ± 0.02 | 16.4 ± 2.4 | 122 ± 4 | 114 ± 4 | 614 ± 26 | 0.16 ± 0.02 | 0.18 ± 0.02 | 11 ± 1 |
| 0.28 ± 0.02 | 1.01 ± 0.01 | 13.0 ± 1.2 | 102 ± 1 | 96 ± 1 | 510 ± 18 | 0.18 ± 0.02 | 0.20 ± 0.02 | 9 ± 1 |
| 0.32 ± 0.02 | 1.00 ± 0.01 | 9.0 ± 1.4 | 94 ± 6 | 89 ± 5 | 435 ± 13 | 0.18 ± 0.03 | 0.19 ± 0.03 | 4 ± 1 |
| 0.37 ± 0.03 | 1.01 ± 0.01 | 10.1 ± 1.2 | 85 ± 3 | 81 ± 2 | 379 ± 10 | 0.17 ± 0.01 | 0.19 ± 0.02 | 9 ± 1 |
| 0.57 ± 0.04 | 0.99 ± 0.02 | 11.1 ± 1.7 | 67 ± 2 | 64 ± 2 | 251 ± 4 | 0.15 ± 0.01 | 0.16 ± 0.01 | 6 ± 1 |
| 0.71 ± 0.05 | 0.99 ± 0.02 | 12.9 ± 1.7 | 57 ± 2.0 | 56 ± 2 | 200 ± 3 | 0.16 ± 0.02 | 0.16 ± 0.02 | 1 ± 1 |
| 0.85 ± 0.06 | 0.98 ± 0.02 | 14.7 ± 2.1 | 51 ± 2 | 50 ± 2 | 167 ± 2 | 0.15 ± 0.06 | 0.15 ± 0.01 | 4 ± 1 |
| 1.04 ± 0.07 | 0.96 ± 0.02 | 15.3 ± 2.0 | 44 ± 2 | 43 ± 1 | 138 ± 2 | 0.15 ± 0.02 | 0.16 ± 0.02 | 5 ± 2 |
| 1.24 ± 0.09 | 0.94 ± 0.02 | 13.0 ± 1.7 | 39 ± 1 | 38 ± 1 | 116 ± 2 | 0.15 ± 0.02 | 0.16 ± 0.02 | 3 ± 1 |
| All | | | | | | 0.16 ± 0.02 | 0.17 ± 0.02 | |



range were not internally mixed with respect to their solute content (Rose et al., 2010; Pöhlker et al. 2018). For particles with $D_{b,a}$ smaller ~ 70 nm (S > 0.6 %) the $MAF_F$ is below one (Fig.8a) dropping to ~ 0.93 and $\sigma_a/D_{b,a}$ is high reaching up to 30-40 % for birch and pine subpollen particles (Fig. 8b). Both results indicate a noticeable portion of externally mixed CCN-inactive particles with much lower

hygroscopicity. As shown in Sect. 3.2 the nebulized aqueous extracts contain a substantial proportion of colloidal particles in the size range of 20-80 nm (Fig. 4). It is therefore possible, that a certain fraction of these particles did not have a sufficient amount of dissolved compounds on the surface, which prevented their activation even at S > 1% (Fig.8a).

        Due to porosity and non-sphericity of the initial dry particles, the shape corrected activation

diameters, $D_a$ (Eq.17) are lower than those initial CCN measured diameters, $D_{b,a}$ (Table 4).

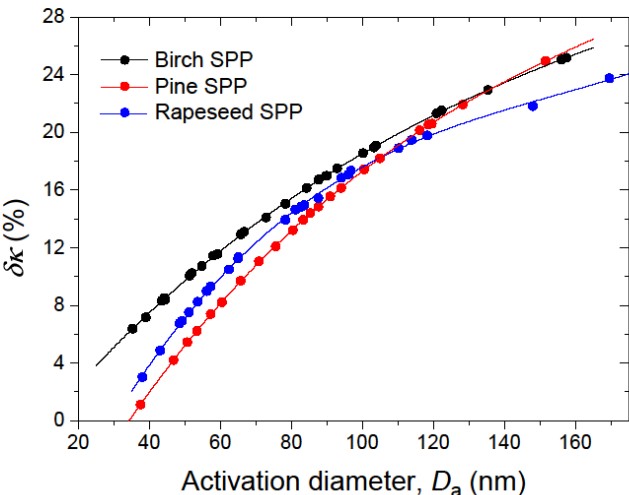

**Figure 9**. The relative change of the hygroscopicity parameter uncertainty, $\delta\kappa$ due to particle irregular structure as a function of activation diameters, $D_a$.   Lines are 3-power polynomial fit to guide the eye.

This difference leads to a significant underestimation of the κ values, especially for large particles (Table 4). Figure 9 illustrates the monotonous growth of $\delta\kappa = (\kappa_a - \kappa_{b,a})/\kappa_a$ with $D_a$ increasing, reaching ~25 % at $D_a$ ~ 160 nm for birch and pine SPP. The shape corrected hygroscopicity, $\kappa_a$ values are shown in Fig. 10. These data demonstrate a weak, but noticeable size-dependent variations in SPP

hygroscopicity. Aerosol particles in the size range of 80-120 nm are more hygroscopic than these in the range from 35 to 80 nm and above 130 nm. The observed variations in $\kappa_a$ are due to the size-dependent ratio of hygroscopic and non-hygroscopic species in the generated dry particles. Most likely, particles with $D_a$ < 60 nm are enriched in primary non- hygroscopic colloids (Fig.4), while particles with $D_a$ > 120 nm may contain their aggregates. The average values of $\kappa_a$, calculated over

the entire supersaturation range, are listed in Table 4 and shown in Fig. 10. For pine, rapeseed and





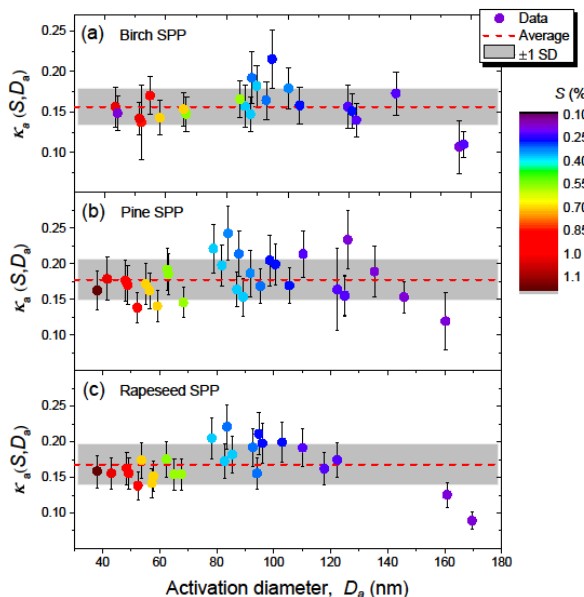

**Figure 10.** Shape corrected hygroscopicity, $\kappa_a$ as a function of aerosol particles activation diameter, $D_a$ and supersaturation, S (color code) of birch (a), pine (b) and rapeseed (c) subpollen particles.

birch, SPP average $\kappa_a$ ($\pm$ SD) is 0.18 $\pm$ 0.03, 0.17 $\pm$ 0.030 and 16 $\pm$ 0.02, respectively. These values correlate quite reasonably with the results of chemical analysis (Table 2), i.e., pine aqueous extracts comprise the highest mass fraction of both ions (0.09) and dissolved organics (0.52). Birch and rapeseed extracts contain a comparable amount of dissolved organics (0.32 and 0.28, respectively), but the content of hygroscopic ions in rapeseed solutions is 1.5 times higher than in birch extracts (0.06 vs. 0.04). In general, the CCN measurement results show that the subparticles of pollen grains at sizes exceeding 45 nm are CCN-active at supersaturations below 1 % (Table 4) and, therefore, can participate in the formation and modification of cloud systems.

## 3.6 Hygroscopic properties

To avoid the uncertainties associated with the aerosol particles morphology, we coupled H&D mode with one of the hygroscopic growth mode, as shown in Fig.1 and previously described in Mikhailov et al. (2020). During the hygroscopic growth experiment the RH2 values in the pre-conditioning section were maintained in the $RH_{\text{H\&D,min}}$ range (Table 3). This range corresponds to $D_{b.\text{H\&D,}min}$ values which are used to approximate the actual mass equivalent diameter of dry particles. Using Eq. (2) the measured RH-dependent mobility particle diameters $D_b$ were converted into hygroscopic growth factors. The experimental growth factors obtained upon hydration and dehydration of preconditioned SPP are illustrated in Fig.11 (panels **a**, **c**, **e**). All samples exhibit gradual and fully reversible water uptake similar to the behavior of amorphous organic substances (Mikhailov et al., 2009; 2013).





**Figure 11**. Growth factors observed in hydration and dehydration experiments (panels **a**, **c**, **e**) and $g_b$-based hygroscopicity parameter, $\kappa_b$ (Eq.13) (panels **b**, **d**, **f**) of birch, pine and rapeseed subpollen aerosol particles with $D_{b,H\&D,min}$ ~ 95 nm, respectively. Different symbols are different experimental runs. Inserts in panels (**a**), (**c**) and (**e**) show measurement uncertainties for different RH ranges. The orange vertical line in panels (**b**), (**d**), (**f**) marks the expected onset of LLPS and the arrow shows the experimental point with the maximum RH value obtained in each measurement.

The fact that the hydration and dehydration curves practically coincide also indicates that there are no kinetic limitations in the water uptake/release on the time scale of the HHTDMA experiment (~7 s). Even at low RH, the growth factor of ~100 nm particles preconditioned at RH ~96 % (dehydration

5    mode) is equal to that obtained in the hydration mode (Table 1). The observed gradual water





uptake/release is determined by the specific chemical composition of SPP comprising a broad spectrum of hydrophilic and hydrophobic organic species as well as inorganic ions (Table 2). Such systems tend to adopt amorphous phase states (Mikhailov et al., 2009; Koop et al., 2011; Shiraiwa et al., 2011). Panels (**a**), (**c**), (**e**) of Fig.11 show that noticeable water uptake by SPP begins at ~30 % RH.

Most likely, above this RH value, the particles undergo a moisture-induced phase transition from glass to semi-solid state that can absorb water and swell (Mikhailov et al., 2009; Koop et al., 2011). Continued absorption observed at RH > 30 % converts the semi-solids particles into heterogeneous solution droplets. At RH > 95 %, the particles take up a significant amount of water, with further RH increasing, particle uptake more, and more water leading to an abrupt increase in the growth factors.

Thus, in case of pine SPP (Fig.11c), a ~1% change in RH from 98 % to 99.2 % results in a 3-fold increase in the particle volume.

Panels (**b**), (**d**), (**f**) of Fig. 11 show the HHTDMA-derived hygroscopicity parameter, $\kappa_b$ as a function of water activity for birch, pine, and rapeseed SPP, respectively calculated from Eq.(13) as described in Sect. 2.8. At the $a_w$ range from ~0.6 to ~0.95 a pronounced decreasing trend of $\kappa_b$ is

observed for all samples. Further growth in water activity ($a_w > 0.95$) leads to a sharp increase in the hygroscopicity parameter. The obtained $g_b(RH)$ and $\kappa_b(a_w)$ dependencies are typical for non-ideal multicomponent systems undergoing liquid-liquid phase separation (LLPS) at high RH.

Modeling results by Renbaum-Wolf (2016) and Rastak et al. (2017) using AIOMFAC (Aerosol Inorganic–Organic Mixtures Functional groups Activity Coefficients), as well as these by Liu et al.

(2018) based on the Flory-Huggins model show the similar humidity-dependent hygroscopicity for the multicomponent organic material. At $a_w < 0.95$, the thermodynamic models predict high $\kappa$ values due to strong interaction among different kinds of molecules that gradually decrease with particle dilution and a sharp increase of $\kappa$ at $a_w > 0.95$ caused by spinodal decomposition resulting in two phases. Above this $a_w$, the formation of an additional phase becomes thermodynamically favorable in

comparison to a single phase. We suggest that consideration of liquid-liquid phase separation, leading to the formation of organic-rich and water-rich phases, can explain the evolution of hygroscopic properties of SPP. Before the liquid-liquid phase separation relative humidity (SRH) is reached, the water uptake represents hygroscopicity of the organic-rich phase, while above SRH, the water uptake is determined by the water-rich phase. Most likely, after LLPS the organic rich-phase includes high

molecular pollen species mainly membrane proteins, starch, and lipids (Stanley and Linskens, 1974) (Table 2). In contrast, in the water-rich phase, the potential components are hydrophilic carbohydrates (glucose, fructose, sucrose, and maltose), water-soluble proteins, and ions (Table 2). Since the analysis of the chemical composition of SPP samples is incomplete and includes only certain classes of compounds, we used Eq. (15) to approximate non-ideal behavior of $\kappa_b$ before phase separation. The



obtained fit curves were extrapolated to $a_w = 1$ and are shown in panels (**b**), (**d**), (**f**) of Fig. 11. The best-fit parameters are listed in Table 5.

**Table 5.** Parameters characterizing the hygroscopic properties of subpollen particles with $D_{b,H\&D,min} \sim 95$ nm: best-fit values (± SD) for the three-parameter fit ($k_1$, $k_2$, $k_3$; Eq. 15) and *SRH* of LLPS; $n$ and $R^2$ are the number of data points and the coefficient of determination of the fit, respectively.

| Species | $k_1$ | $k_2$ | $k_3$ | $R^2$ | $a_w$ range | $n$ | SRH (%) |
|---|---|---|---|---|---|---|---|
| Birch SPP | 0.356 ± 0.310 | -0.310 ± 0.079 | 0.109 ± 0.049 | 0.937 | 0.55 − 0.97 | 66 | 97.7 |
| Pine SPP | 0.269 ± 0.031 | -0.041 ± 0.077 | -0.084 ± 0.048 | 0.953 | 0.55 − 0.96 | 71 | 98.2 |
| Rapeseed SPP | 0.291 ± 0.040 | -0.329 ± 0.108 | 0.189 ± 0.072 | 0.672 | 0.55 − 0.93 | 31 | 94.6 |

The difference between extrapolated and measured $\kappa_b$ demonstrates the effect of LLPS on the water uptake. The onset $a_w$ of LLPS was determined as a minimum in the $\kappa_b(a_w)$ dependency ($d\kappa_b/da_w = 0$) and corresponding SRH values converted from obtained $a_w$ of LLPS are given in Table 5. The onset SRH of the birch SPP (97.7 %) and pine SPP (98.2 %) are comparable, while the SRH of rapeseed SPP is remarkably lower (94.6 %). The SRH value depends on the dipole moment of organic material, and this parameter is a function of the elemental oxygen to carbon (O:C) ratio. Multiple experiments with organic material showed that LLPS has been observed for O:C < 0.5 and never observed for O:C > 0.8, and this rule applies to both, simple and complex mixtures (You et al., 2014). Parameterizations reported by Bertram et al. (2011) and Song et al. (2012) indicate that the SRH tends to decrease with an increasing O:C ratio. It is possible, therefore, that the lower SRH obtained for rapeseed subparticles is caused by a higher content of more oxidized organic compounds relative to birch and pine samples.

The main water active organic components of SPP are carbohydrates and proteins (Table 2). For carbohydrates the O:C ratio is close to one, therefore, it is unlikely that they can initiate LLPS. The average O:C ratio in protein-forming amino acids is 0.4; thus, proteins are suitable candidates for triggering spinodal decomposition. The SRH at that O:C ratio is in the range of 95-100% with a weak effect of organic-inorganic ratio on SRH (You et al., 2014; Bertram et al. 2011). The experimental SRH values for SPP are within 95-98%, indicating that proteins as well other organic compounds with an O:C ratio near 0.4 may be responsible for LLPS.

Dissolution of sparingly soluble organic compounds may also contribute to the hygroscopicity. Modified $\kappa$-Köhler models that assumes limited water solubility predicts a monotonic increase of $\kappa$ values for an $a_w$ increasing (Petters et al., 2009; Pajunoja et al., 2015). However, the decreasing trend of $\kappa_b$ observed in the $a_w$ range of 0.6-0.95 cannot be explained by these models (panels (**b**), (**d**), and (**f**) of Fig. 11). In addition, the solubility-based prediction is inconsistent with the size-resolved CCN



measurements. As can be seen from Table 4, the growth factor ($g_a = D_{wet,a}/D_a$) at the activation point increases with an increasing $D_a$ (decreasing $S$) that should results in more organic solute dissolved in the aqueous phase, thus increasing the $\kappa$ value. The CCN-derived $\kappa_a$ values rather demonstrate the opposite effect (Fig.10), that is, an increase in $D_a$ on average leads to a decrease in

$\kappa_a$. Most likely, this dependence results from the unequal composition of the dry particles, as discussed above. Taking into account the above arguments, we tend to assume that sparingly soluble compounds have little effect on the water uptake of the SPP studied.

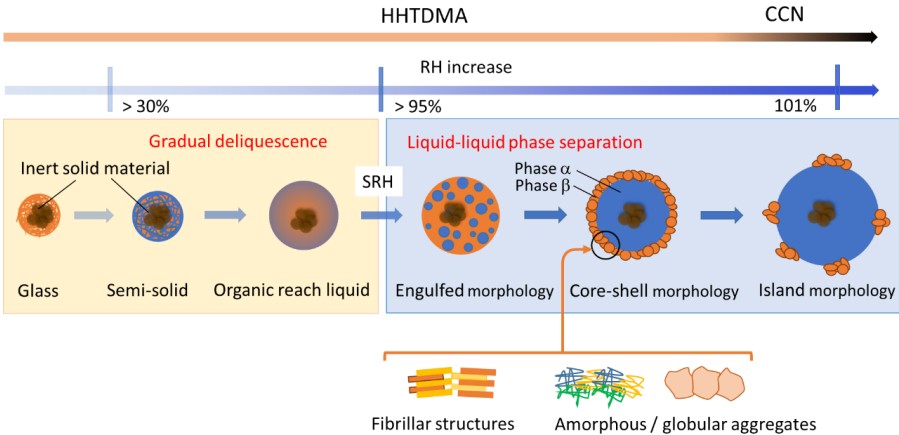

**Figure 12.** Water uptake by amorphous subpollen particles: possible morphology and processes. SRH is the RH onset of liquid-liquid phase separation.

Figure 12 shows the possible evolution of SPP morphology upon interaction with water vapor.
This cartoon unites early studies with amorphous particles (Mikhailov et al., 2009; Koop et al., 2011) and those relatively new results associated with LLPS of multicomponent systems (Renbaum-Wolff et al., 2016; Ruehl et al., 2016; Song et al., 2017a; Ovadnevaite et al., 2017; Rastak et al., 2017). At low RH particles are in an amorphous glassy state. Due to the small diffusion coefficient of water ($D_{H_2O}$) in the glassy matrix (typically below $10^{-10}\,\mathrm{cm^2\,c^{-1}}$ at 300 K) the water uptake occurs only at the
particle surface. Upon further increase in RH, the outermost layers will absorb more water, which softens the matrix and act as plasticizer, thus reducing viscosity and increasing $D_{H_2O}$. At RH range above ~30%, particle undergo a humidity-induced phase transition from glass to a semi-solid state. This induces a self-accelerating process such that significant water uptake occurs in a gradual deliquescence process, and the semi-solid turns into organic-rich liquid. At high RH (> 95%), particles
containing hydrophobic and hydrophilic organics undergo LLPS forming water-rich ($\alpha$) and organic-rich ($\beta$) phases (Fig.12). The specific value of SRH is a function of O:C ratio and ions content (salting in/out effects) (You et al., 2014, and references therein). On the initial stage of LLPS particles, the partially engulfed morphology is preferable; with a further RH growth, the water-enriched inclusions





increase in size and coagulate into one phase forming particles with core-shell morphology; the aqueous phase being mostly responsible for particle growth at high RH. Under continuing growth condition, the organic shell will reach a relatively small volume compared to the growing volume of the aqueous core phase, such that it becomes impossible to form a complete organic shell. Under such

conditions, the organic material may spread out as molecular "islands" at the surface of the core (Fig.12) (Ovadnevite et al., 2017). The transition from a coherent phase (organic film) to a spread out "gaseous" state on a 2-dimensional (2D) droplet surface can be modelled as a 2D phase transition, as described, by Ruehl et al. (2016). The presented particle morphology after LLPS (Fig.12) is based on microscopy images of supermicron samples (Renbaum-Wolff et al., 2016; Song et al., 2017a) and can

differ from ~100 nm particles studied here. Our results, however, are in agreement with these images. Panels (**b**), (**d**), and (**f**) of Figure 11 clearly show that at SRH, water uptake does not occur instantaneously, but rather within a certain $\Delta RH$, suggesting that a phase separation includes a sequential morphological transformation  considered above. A similar agreement between microscopic images and the phase transition processes govern the water interactions by the 100 nm

particles was observed by Rastak et al. (2017) in the case of monoterpenes derived SOA.

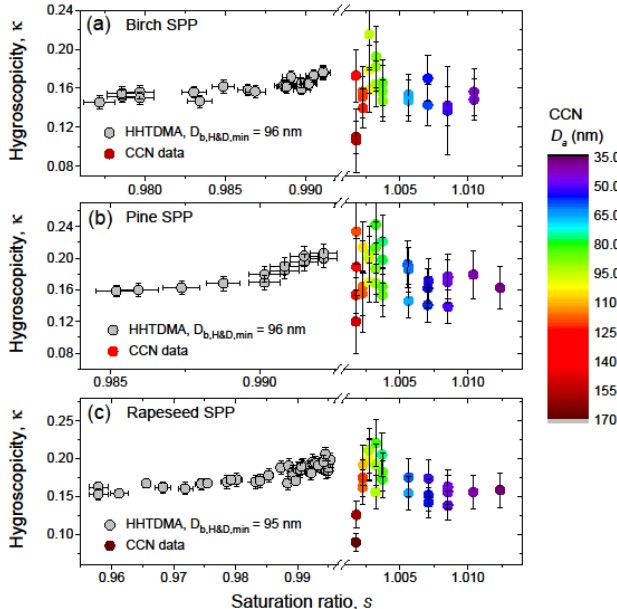

**Figure 13.** HHTDMA-based and CCN-based hygroscopicity parameter, $\kappa$ for birch (**a**), pine  (**b**) and rapeseed (**c**) SPP as a function of RH. The color code shows the CCN-derived $\kappa_a$ with respect to the shape corrected activation diameters, $D_a$ .



Figure 13 shows the hygroscopicity parameter, $\kappa$ measured by HHTDMA and CCNC in SPP. The HHTDMA-derived $\kappa_b$ values taken after $a_w$ of LLPS (orange line in panels (**b**), (**d**), and (**f**) of Fig.11). One can see that within the measurement uncertainty of CCN setup, the maximum values of the HHTDMA-derived $\kappa_b$ with $D_{b,H\&D,min}$ ~ 96 nm coincide with the CCN-derived $\kappa_a$ with $D_a$ ~100 nm (yellow data points) having a similar chemical composition, i.e., 0.17±0.01 vs. 0.18±0.03; 0.20±0.01 vs. 0.21±0.03; and 0.19±0.01 vs. 0.20±0.03 for birch, pine, and rapeseed SPP, respectively. Note the agreement in κ is achieved assuming that the surface tension equals $\sigma_w$, suggesting that the effect of the organic coating is negligible. One possible explanation is that the organic material establishes a partial shell covering of the water-rich phase ("island" morphology, Fig.12). Following to Ovadnevaite et al. (2017) the surface coverage parameter, $c_\beta$, is

$$c_\beta = min\left[\frac{V_\beta}{V_\delta}, 1\right], \tag{18}$$

where $V_\beta$ is the volume of phase $\beta$ and $V_\delta$ is the volume of a spherical shell of minimum thickness $\delta_{\beta,min}$, corresponding to a molecular single layer. In this study the volume of phase β was calculated as

$$V_\beta = \frac{\pi}{6}D_{H\&D,min}^3 \times 0.07, \tag{19}$$

assuming that the hydrophobic organic volume fraction in the dry particles is ~7%. This assumption is based on the chemical analysis of aqueous pollen extracts (Table 2), containing high molecular species such as protein ~3%, extractable starch ~2%, and other uncounted macromolecules ~2% (lipids, phenolic compounds, etc.). Here we assume that mass and volume fractions of pollen species are equal. Inert (surface inactive) solid material, such as cross-linking callose, sporopollenin, and cellulose is not considered (Sect. 3.1). The volume of spherical shell was derived from

$$V_\delta = \frac{4}{3}\pi\left[\left(\frac{D_{wet}}{2}\right)^3 - \left(\frac{D_{wet}}{2} - \delta_{\beta,min}\right)^3\right], \tag{20}$$

where $\delta_{\beta,min}$ is used as a free parameter varying from 0.4 to 5 nm. In the previous studies (Ruehl et al., 2016; Renbaum-Wolff et al., 2016; Liu et al., 2018) the $\delta_{\beta,min}$ value of 0.16 and 0.3 nm was used for LLPS modeling of multicomponent organic mixtures with molar mass (*M*) ranging 150-360 Da (Zuend and Seinfeld, 2012). The molar mass of biomolecules and their size is significantly larger. In particular, the *M* of plant proteins is mainly 0.5-500 kDa (Mohanta et al., 2019), and their volume-equivalent diameter is 2.2 -10.4 nm, respectively (Erickson, 2009); therefore, we expanded the possible range of $\delta_{\beta,min}$ up to 5 nm. The effective surface tension of the droplet, σ is calculated as the weighted mean of the composition-dependent surface tensions from α and β phases (Fig. 12):

$$\sigma = (1 - c_\beta)\sigma_\alpha + c_\beta\sigma_\beta, \tag{21}$$




where $\sigma_\alpha$ and $\sigma_\beta$ is the surface tension of the water-rich and organic-rich phase. Figure 14 shows the calculation results for $D_s = 100$ nm based on Eqs. (18)-(20), which presented as $c_\beta(g = W_{wet}/D_s)$

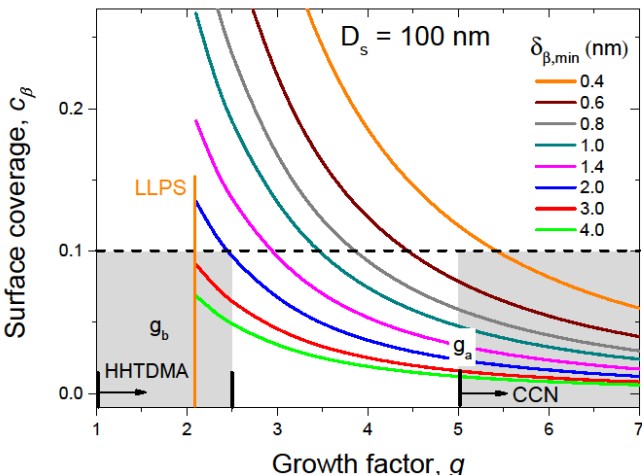

**Figure 14.** The surface coverage, $c_\beta$ as a function of particle growth factor, $g = D_{wet}/D_s$ with $D_s = 100$ nm, obtained for different values of the monolayer thickness, $\delta_{\beta,min}$ (colored lines). For HHTDMA and CCN measurements, $D_s$ equals $D_{H\&D,min}$ and $D_a$, respectively. $g_a = D_{wet,a}/D_a$ is the growth factor at the activation diameter, $D_a \approx 100$ nm. The gray areas show the growth factor range of HHTDMA and CCN measurements. The dashed line indicates the $c_\beta$ value bellow which $\sigma \approx \sigma_w$. Both $g_a$ and $g_b$ growth factors are average values of the birch, pine, and rapeseed samples.

dependences, obtained for different values of the monolayer thickness. One can see that at $\delta_{\beta,min} > 2$

5    nm both for the HHTDMA- and CCN-derived growth factors obtained after LLPS, the surface coverage does not exceed 0.1. Using $c_\beta = 0.1$ and assuming that the surface tension of water-rich and organic-rich phase is 72 and 57 ± 5  mNm$^{-1}$ (Absolom  et al., 1981;  Makievski et al., 1990; Alahverdjieva et al., 2008), respectively, it follows from Eq. (21) that σ equals 70.5 ± 0.5  mNm$^{-1}$. That is, for particles with $D_s = 100$ nm at $\delta_{\beta,min} > 2$ due to partial shell covering (Fig.12, "island"

10    morphology), the surface effect of biomolecules becomes negligibly small. A small contribution of organic coating to the surface tension of the diluted droplets was also observed in CCN activation experiments with pollenkitt particles (Prisle et al., 2019).

In our simplified calculations, protein and extractable starch were chosen as a main surface-active components. Depending on polarity, hygroscopicity, concentration, and electrostatic charge,

15    this group of biomolecules  can form higher-order fibrillary, globular, and amorphous aggregates, significantly affecting the thermodynamic and kinetic properties of the  surface layer (Fig.12) (Lad et al., 2006; Jarpa-Parra, et al., 2015; Zapadka et al., 2017; Trainor et al.,2017; D'Imprima et al., 2019).



Moreover, some biopolymers are initially prone to forming two-dimensional aggregates at the air/water interface, making the assumption of mono/multi-layer adsorption redundant (Fainerman and Miller, 1999). In addition, other organic and inorganic species present in the SPP may effect on the thermodynamic properties of the air/water interface, decreasing or increasing surface tension of the droplets (Lin and Timasheff, 1996; Fainerman et al 1998; Kotsmar et al., 2009). Additional studies are needed to specify the effect of biomolecules on the aerosol particles hygroscopic and CCN properties, including LLPS stage.

## 4  Summary and conclusions

In this study, we have presented measurement results of hygroscopic and cloud condensation nuclei properties of biological aerosols obtained by nebulization and drying of the birch, pine, and rapeseed pollen aqueous extracts. These studies were supplemented with chemical analysis of the pollen species, as well as gravimetric and dynamic light scattering (DLS) measurements to determine the mass fraction of hydrosols and their size distribution, respectively. The analysis results suggest that the initial dry particles in the size range of 20-190 nm are a mixture of different proportions of water-soluble and water-insoluble compounds. The analysed fraction of water-extractable biomolecules includes proteins and starch. Among the water-soluble compounds, the main ones are inorganic ions and monosaccharides. Aqueous extracts contain also a significant amount of water-insoluble species (~50 wt %) which are mainly in the size range of 20-80 nm.

The HHTDMA studies in the H&D (restructuring) mode have shown that the DMA1 selected particles ranging 30-180 nm  have a porous structure wherein the void fraction gradually increases with size, reaching ~26% for 180 nm birch and pine SPP. The minimum mobility diameters obtained in H&D mode were used as a proxy of the mass equivalent diameters of the initial dry particles.

CCN measurements revealed that the shape corrected dry activation diameters measured at $S$ = 0.18-1.24 % are in the range of 137-44 nm, 134-38 nm, and 159-38 nm for birch, pine and rapeseed SPP, respectively. For all samples at high $S$ (> 0.6%), the maximum activated fraction is generally well below one, and heterogeneity range ($\sigma_a/D_{b,a}$) reaching up to 30-40 %. Both indicate a substantial portion of externally mixed CCN-inactive particles with much lower hygroscopicity – most likely colloids, which initially presents in the nebulized solution. A weak, but noticeable size-dependent variation in $\kappa$ was observed, ranging from 0.13-0.18; 0.16-0.20 and 0.12-0.20 for birch, pine and rapeseed SPP, respectively. For all SPP studied the maximum $\kappa$ values refer to $D_a$ = 80-120 nm, and low values correspond to particles with activation diameters below and above this size range. Most likely, particles with $D_a$ < 80 nm are enriched in primary non-hygroscopic colloids, while particles with $D_a$ > 130 nm may contain their aggregates.




The sensitivity analysis of the CCN results has shown that the uncertainty in hygroscopicity parameter, $\delta\kappa$ caused by irregular particle morphology is strongly dependent on their size, increasing from 4 % to 25 % with particle size increases from 30 nm to 160 nm, respectively. In general, the CCN measurement results show that the SPP at sizes exceeding 45 nm are CCN-active at

supersaturations less than 1 % and, therefore, can participate in the formation and modification of cloud systems.

The hygroscopic properties of SPP with $D_{H\&D,min}$~ 96 nm were investigated by the high humidity tandem differential mobility analyser (HHTDMA) ranging 2 - 99.5 % RH. Over the entire RH range, the water uptake of SPP occurred gradually and reversibly, similar to the behavior of

amorphous organic substances. A significant amount of absorbed water ($g_b$ >1.4) was observed at RH > 95 %, with further RH increasing (RH > 97%), particle uptake more water leading to an abrupt increase in the growth factors: approximately ~1% RH results in a 3-fold increase in the particle volume. Due to the solution non-ideality, the HHTDMA-derived hygroscopicity parameter, $\kappa$ exhibits a decreasing trend in the 0.6-0.95 $a_w$ range and sharp increases at $a_w > 0.95$. We suggest that the

observed $g(RH)$ and $\kappa_b(a_w)$ dependences at high RH are the result of liquid-liquid phase separation of the multicomponent SPP. For birch, pine and rapeseed subpollen particles, the onset RH of LLPS was estimated to be 97.7%, 98.2%, and 94.6%, respectively.

A good agreement in κ was obtained between HHTDMA at RH > 97 % and CCN-derived values. We believe that this closure became possible by the extended upper limit in RH of our

HHTDMA system (up to 99.5%). Discrepancy between the κ, measured below and above water saturation in the organic and organic-inorganic mixed particles have been reported in several studies (Song et al., 2017a; Zhao et al., 2016; Hansen et al., 2015; Pajunoja et al., 2015; Whitehead et al., 2014; Alfarra et al., 2013; Dusek et al., 2011; Massoli et al., 2010, Good et al., 2010). Petters et al. (2006), Hodas et al. (2016). Renbaum-Wolf et al., (2016), Rastak et al. (2017), and Liu et al. (2018)

suggested that such discrepancies are expected in systems that undergo LLPS at high RH. In conventional HTDMA systems, the upper RH limit rarely exceeds 95% (Mikhailov et al., 2020). Usually, the CCN-derived κ values are compared with those obtained by the HTDMA at RH < 95 %, which leads to their discrepancy, especially for the organic particles with low O:C ratio (Song et al., 2017a). In this study, the growth factors were also measured at RH above LLPS so that both HHTDMA

and CCN measurements provides the κ values of the water-rich phase, which enabled us to close the gap between the particle's hygroscopic growth and their CCN activation. The closure between HHTDMA- and CCN-derived κ was achieved when the value of σ was that of water, suggesting that above LLPS, due to large growth factors, the surface-active biomolecules are not able to fully coat the water-rich phase. Most likely, an "islands" composed of 2D or 3D aggregates at the air-water interface

are forming.



An important question, when considering the role of SPP in atmospheric processes is the high uncertainty in the number concentration produced by a pollen grain ($n_{spg}$). SEM analysis by Suphioglu et al. (1992) showed that when the ryegrass pollen grain is immersed in water it release up to 700 starch granules in the range of 0.6–2.5 µm. Controlled environment chamber experiments by Taylor et al. (2002) with flowering stems of ryegrass indicated that ruptured pollen release SPP in smaller size fractions (0.1–1µm) and more particles than the starch granules previously reported by Suphioglu et al. (1992). By combining the SPP size distribution of Taylor et al. (2002) with the SEM results of Suphioglu et al. (1992) it follows that the number fraction of SPP in 0.1-1 µm size range per pollen grain is about $10^4$. Note that the SPP size distribution described in Taylor et al. (2002) was measured by optical counting and therefore did not account for bioparticles smaller than 0.1 µm. The upper limit of $n_{spg}$ can be obtained if all pollen mass converted to SPP. Our estimate based on SPP size distribution with mode $D_s = 100$ nm and SPP density, $\rho \sim 1$ g m$^{-3}$ yields $n_{spg} = 10^6$ -$10^7$ (Mikhailov et al., 2019). A close value of $n_{spg} = 10^6$ was obtained by Wozniak et al. (2018), assuming $D_s = 200$ nm and $\rho = 1.4$ g m$^{-3}$. Most likely, the real values of $n_{spg}$ are lower than their upper estimate. Thus, atmospheric measurements by Hader et al. (2014) showed no correlation between ambient pollen grain counts and observed IN concentrations during the pollination season. Additional laboratory and atmospheric studies are required to clarify the number of SPP released by pollen under different moisture conditions.

In conclusion, the obtained and discussed hygroscopic and CCN properties of the SPP, including spinodal decomposition, may be common for atmospheric particles, such as sea-spray aerosols (Ovadnevaite et al., 2011; Estillore et al., 2017; Lee et al., 2020) as well as natural and anthropogenic aerosols, containing a broad spectrum of biopolymers coupled with water-soluble organic and inorganic material in different proportions (Xu et al., 2020; Song et al., 2017b; Fröhlich-Nowoisky et al., 2016; Pöschl and Shiraiwa, 2015; You et al., 2014; Pöhlker et al., 2012; Schneider et al., 2011; Graber and Rudich, 2006; Zhang and Anastasio, 2003). In addition, the ability of SPP undergo LLPS at subsaturated conditions forming a water-rich phase may enhance post-translational modifications of the proteins induced by air pollutants (O₃, NO₂) and therefore reinforce their allergic potential. This information may be useful for prognostic assessments of allergic diseases such as thunderstorm asthma (Suphioglu, 1998; Pöschl et al., 2015; Reinmuth-Selzle et al., 2017; Beggs, 2017; Kornei, 2018; Yair et al., 2019, Hughes at al., 2020).

*Data availability.*

*Supplement.*

*Author contributions.* EFM designed the study, performed the concomitant measurements, and wrote this paper, MLP, LAK, and OOK conducted the CCN measurements, KR-S and JF-N conducted the chemical analyses, AAK., OAI, SSV, CP, and UP contributed to the discussion of the results.



*Competing interests.* The authors declare that they have no conflict of interest.

*Acknowledgements.* We thank the Geomodel Research Center, Interdisciplinary Resource Center for Nanotechnology, Chemical Analysis and Materials Research Centre and Center for Optical and Laser Methods of Matter Research at Saint Petersburg State University. We also thank Mohammed Katzia for technical support in the course of the CCN measurements.

*Financial support.* This research has been supported by the Russian Science Foundation (grant no. 18-17-00076) and Max Planck Society.

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
