# Peer review of "Water uptake of subpollen aerosol particles: hygroscopic growth, CCN activation, and liquidliquid phase separation"

_Atmospheric Chemistry and Physics, 2020_

## Short Comment (SC1) · 2 Dec 2020

We have published another paper related to pollen hygroscopicity (Chen et al., 2019), which may be relevant for the manuscript submitted by Mikhailov et al. (2020).

Chen, L. X. D., Chen, Y. Z., Chen, L. L., Gu, W. J., Peng, C., Luo, S. X., Song, W., Wang, Z., and Tang, M. J.: Hygroscopic properties of eleven pollen species in China, ACS Earth Space Chem., 3, 2678-2683, 2019.

---

## Referee Comment (RC1) · Mingjin Tang (Referee) · 1 Feb 2021

I would like to thank the editor for inviting me to review this manuscript. Mikhailov et al. used several methdos to investigate composition, hygrosocpic properties and CCN activities of three types of subpollen aerosol particles. This work is very robust and comprehensive, and thus deserves publication by ACP. This work involves lots of knowledge in aerosol microphysics and is thus quite beyond my expertise; the editors may need to ask for additional reviews. 1) In general I feel that this manuscript is very long, and the authors may considering moving some non-critical parts to Appendix or Supporting Information. Here I provide a few examples: ii) Section 2.5 (including

[Figure]

Figures 1-2 and Table 1) can be substantially shorten, as HHTDMA has been described in a previous paper; some mathematical equations (and related discussion) presented in in Sections 2.6 and 2.8 can be moved to Appendix or Supporting Information. ii) p35, line 1-18: I am not sure whether information provided by this long parapraph is an important finding of this work. 2) p27, line 14-15: could you please explain why the measured kappa values decrease with water activity in the range of 0.65-0.95 before LLPS occurs? 3) p 25, line 2: please change "16" to "0.16".

---

## Referee Comment (RC2) · Anonymous Referee #2 · 11 Feb 2021

Eugene et al presented a set of comprehensive hygroscopicity measurement of subpollen particles. The paper is well written, and the experimental results are robust. The author measure hygroscopicity ranged from 2 to 1.2%, and the liquid-liquid phase separation model was used to explain the solution non-ideal behaviour. The literature review in the introduction is well written and informative. This is an interesting study and should be published in ACP with minor revisions.

Here are some detailed comments:

The paper is long and hard to follow, there are few parts not necessarily to be in the main text, e.g. section 3.1, 3.2 and 3.4.

[Figure]

Page 2 line 33: The reference of kappa is missing.

Figure 1 and 2 can go to supporting information.

page 14 section 3.1. is there any size-resolved chemical analysis? I assume figure 3 presented the bulk chemical compositions.

Page 16. It is not clear why the hydrodynamics size distribution and the aging experiments were performed? What is the implication of this experiment?

Page 21 Figure 7. What is the difference between a1 a2 a3 a4? I assume it is the different pictures of the same sample.

Figure 7. The description of SEM can go to the Method section

Page 25 line 3: 0.16 not 16.

Page 25 line 4: Any calculation of kappa based on chemical composition?

Page26 Figure 11: can you added the shaded area in the left column of GF-RH curves using kappa model, with the lowest and highest kappa in the right column.

Figure 13. Is the difference between Brich SPP, Pine SPP and Rapeseed SPP statistical significant?

Figure 14. Is the liquid-liquid phase separation model used to reproduce the observations? If yes, is it comparable?

Page 34 line 24 it is hard to know Hods et al. (2016) belongs to which sentence.

Page 35 line 1. To me this paragraph is not really related to the main topic of this paper.

Page 35 line 21, is sea-spray aerosol a kind of natural aerosol?
* * *

---

## Author Comment (AC1) · 23 Mar 2021

Thanks a lot! We used Chen et al., 2019 paper in the main text and the Reference list.

---

## Author Comment (AC2) · 23 Mar 2021

Response to referee Mindjin Tang

The referee's comments are in italics, our responses in plain font.

Mikhailov et al. used several methods to investigate composition, hygroscopic properties and CCN activities of three types of subpollen aerosol particles. This work is very robust and comprehensive, and thus deserves publication by ACP. 1) In general I feel that this manuscript is very long, and the authors may considering moving some non-critical parts to Appendix or Supporting Information.

We thank Mindjin Tang for the constructive criticism and suggestions for improvement that were taken into account upon manuscript revision. Responses to individual comments are given below.

1. Section 2.5 (including Figures 1-2 and Table 1) can be substantially shorten, as HHTDMA has been described in a previous paper; some mathematical equations (and related discussion) presented in Sections 2.6 and 2.8 can be moved to Appendix or Supporting Information.

Figures 1-2 and Table 1 went to Supporting information. Sections 2.6 and 2.8 were substantially shorten. Some equations and accompanying material sent to the Supplement.

2. p35, line 1-18: I am not sure whether information provided by this long parapraph is an important finding of this work.

This paragraph has been removed.

**3. p27, line 14-15: could you please explain why the measured kappa values decrease with water activity in the range of 0.65-0.95 before LLPS occurs?**

Kappa value decreases with  $a_w$  increasing due to solute-solute and solute-water interaction in the concentrated solution droplet. It particularly follows from relation for intrinsic hygroscopicity:  $\kappa_{intr} \approx v_s \Phi_s \frac{\rho_s M_w}{\rho_w M_s}$ , where  $v_s \Phi_s$  is the product of the stoichiometric dissociation number and the molal osmotic coefficient of the solute. For concentrated solutions  $\Phi_s > 1$ , in dilute solution it approaches to 1. There are different thermodynamic models (like UNIFAC) used to describe the concentration effect of organic on  $\Phi_s$ . Nonideality of concentration solution also considers in term of the excess of Gibbs energy and water activity coefficient,  $\gamma_w$  (Petters et al., Atmos. Chem. Phys., 9, 3999–4009, 2009. Concentration-dependent of  $\kappa$  for single compounds in term of van't Hoff factor ( $i_s$ )  $\kappa_{intr} = i_s \frac{\rho_s M_w}{\rho_w M_s}$ , analysed in Mikhailov et al. (Atmos. Chem. Phys., 9, 9491–9522, 2009) Mikhailov et al. (Atmos. Chem. Phys., 13, 717–740, 2013) also suggested a mass-based  $\kappa$  interaction model that describes concentration-dependent water uptake by multicomponent aerosols.

4. *p 25, line 2: please change "16" to "0.16".* Done.

---

## Author Comment (AC3) · 23 Mar 2021

Response to referee #2

The referee's comments are in italics, our responses in plain font.

*Eugene et al presented a set of comprehensive hygroscopicity measurement of subpollen particles. The paper is well written, and the experimental results are robust. The author measure hygroscopicity ranged from 2 to 1.2%, and the liquid-liquid phase separation model was used to explain the solution non-ideal behaviour. The literature review in the introduction is well written and informative. This is an interesting study and should be published in ACP with minor revisions.*

We thank referee #2 for the constructive criticism and suggestions for improvement that were taken into account upon manuscript revision. Responses to individual comments are given below.

1. *The paper is long and hard to follow, there are few parts not necessarily to be in the main text, e.g. section 3.1, 3.2 and 3.4.*

   These sections are essential parts of the paper. They contain results that are used to interpret the CCN and HHTDMA measurements. In my opinion, it is inconvenient to repeatedly refer to the Appendix or Supplement to find the desired information. Alternatively, Sections 2.6 and 2.8 were substantially shorten. Some equations and accompanying material we sent to the Supplement.

2. *Page 2 line 33: The reference of kappa is missing.*

   The reference to Petters and Kreidenweis, 2007 is added.

3. *Figure 1 and 2 can go to supporting information.*
   Done.

4. *page 14 section 3.1. is there any size-resolved chemical analysis? I assume figure 3 presented the bulk chemical compositions.*

   Yes, it is. We clarified this point: "**Bulk** chemical analysis results of water-extractable compounds are summarized in Table 1 and illustrated in Fig. 1."

5. *Page 16. It is not clear why the hydrodynamics size distribution and the aging experiments were performed? What is the implication of this experiment?*

   As shown in Sect.3.1 and underlined in Sect.3.2, the dry aerosol is an external mixture of water-soluble and water-insoluble compounds. The effect of the water-insoluble compounds (hydrosols) on particle hygroscopicity depends on its size range. DLS measurements (Sect.3.2) show that the size of colloidal particles spans the range of 40-110 nm i.e. within the SPP size range of dry particles (20-190 nm) used for CCN and HHTDMA measurements. This information was further used to characterize size-dependent CCN properties of SPP (Sect.3.5).
   As noted in Sect.3.2, the aging of nebulized solution caused by hydrosols coagulation can change the size-selected dry particle composition during the HHTDMA (8 h) and CCN (4 h) experiments. DLS and turbidity test measurements showed that this effect is insignificant in the specified time periods, while it becomes noticeable after 10 h of aging (Suppl. S6).

6. *Page 21 Figure 7. What is the difference between a1 a2 a3 a4? I assume it is the different pictures of the same sample.*

   You are right. To avoid confusions we have only saved one image for each sample.

7. *Figure 7. The description of SEM can go to the Method section.*

Done.

8. *Page 25 line 3: 0.16 not 16.*

   Corrected.

9. *Page 25 line 4: Any calculation of kappa based on chemical composition?*

   To calculate $\kappa_{chem}$ at least concentrations of organic molecules and neutral salts have to be known. Our chemical analysis is not full. For example, we know total concentrations of water-soluble carbohydrates an proteins but not their molecular composition. Neutral salts concentrations are also needed for kappa calculation. Unfortunately, ion balance of anions and cations does not converge. The most likely cause is the ability of charged proteins to bind cations and anions selectively (salting in effect). For the above reasons, we did not compute $\kappa_{chem}$.

10. *Page 26 Figure 11: can you added the shaded area in the left column of GF-RH curves using kappa model, with the lowest and highest kappa in the right column.*

    The κ-Kohler curves with min. and max. kappa values are shown as inset in Fig.9 (left panels). The following accompanying text was added:
    "Insert in panels (**a**), (**c**), (**e**) of Fig. 9 (old Fig.11) shows the κ-Köhler modeling results (Eq.6) with minimum and maximum $\kappa_b$ values observed on the $\kappa_b(a_w)$ dependences (panels (**b**), (**d**), (**f**) of Fig.9). The difference in the Köhler curves reflects potential uncertainty arising from spinodal decomposition. The lower Köhler curve fits the particle growth factors before onset SRH (orange line), while upper Köhler curve corresponds to maximum $\kappa_b$ observed after LLPS. Intermediate data points indicate gradual phase separation accompanied by the particle engulfed morphology (Fig.10)".

11. *Figure 13. Is the difference between Brich SPP, Pine SPP and Rapeseed SPP statistical significant?*

    As outlined in Sect. 3.5 and shown in Fig.10 (now Fig. 8), the difference between Birch SPP, Pine SPP, and Rapeseed SPP in the κCCN values averaged throughout the 35-170 nm size range is not statistically significant. However, noticeable size-dependent variations in SPP hygroscopicity are traced. Aerosol particles in size range of 80-120 nm are more hygroscopic ($\kappa_a \sim 0.20$) than these in the range from 35 to 70 nm ($\kappa_a \sim 0.14$) and above 130 nm ($\kappa_a \sim 0.13$). The observed variations in $\kappa_a$ are due to the size-dependent ratio of water-soluble and water-insoluble material in the dry particles. We slightly polished the text in Sect.3.5 to specify the difference between mean and size-dependent kappa.

12. *Figure 14. Is the liquid-liquid phase separation model used to reproduce the observations? If yes, is it comparable?*

    Not at the moment, but I am pondering this question. In the case of AIOMFAC or E-AIM model, the quantity of subgroups has to be known. It is a problem since molecular composition of SPP is not known. We assume that proteins with C/O ~0.4 trigger LPPS, but their thermodynamic properties are poorly defined within existing models. Particularly, peptide bond (CHON) is not present in the subgroup list. Besides, effects of intramolecular interactions between functional groups are usually not considered in AIOMFAC and E-AIM models. Recent results by Luo et al. (Sci. Total Environ., 734 (2020) 139318) showed that even for single amino acids, the measured growth factors were much lower than those predicted by the E-AIM using standard UNIFAC model. We plan to model LLPS using the Flory-Huggins model, which is more suitable for mixed solutions containing water-soluble polymers.

13. *Page 34 line 24 it is hard to know Hods et al. (2016) belongs to which sentence.*

Corrected

14. *Page 35 line 1. To me this paragraph is not really related to the main topic of this paper.*

    This paragraph has been removed.

15. *Page 35 line 21, is sea-spray aerosol a kind of natural aerosol?*

    Text corrected.

    I would like to thank the reviewer#2 again for helpful and constructive comments.